# Physics-informed Neural Operator Learning for Nonlinear Grad-Shafranov Equation

**Siqi Ding**[1]  **Zitong Zhang**[2][3]  **Guoyang Shi**[1]  **Xingyu Li**[4]  **Xiang Gu**[1]  **Yanan Xu**[1]  **Huasheng Xie**[1]
**Hanyue Zhao**[1]  **Yuejiang Shi**[1]  **Tianyuan Liu**[1]

## Abstract

AI for fusion requires bridging a critical "sim-to-real" gap: simulation-trained models must generalize reliably under distribution shifts in safety-critical workflows. Focusing on the nonlinear Grad-Shafranov equation (GSE), we develop and analyze a physics-anchored operator-learning framework for fixed-boundary equilibrium prediction. The framework combines data anchors with PDE residual constraints and uses a physics-motivated Transformer-KAN Neural Operator (TKNO) to capture global elliptic coupling and nonlinear source response. Under multi-parameter distribution shifts, our analysis shows that data-only surrogates can develop severe OOD tails, while physics-only training may converge to incorrect solution branches; by combining sparse data anchors with PDE constraints, physics-anchored training reduces worst-tail errors on shape-driven and joint shifts. Mechanism diagnostics indicate that data anchors suppress branch-scale failures, whereas PDE residuals reduce OOD-induced high-frequency error amplification. Evaluated on EXL-50U discharge inputs against the device's operational equilibrium solver, the model achieves close agreement (RMSE < 1.3%) with millisecond-level inference. These results provide a practical route toward physically reliable AI surrogates for fusion workflows. Our code will be available at https://github.com/dsqzhou/physics-anchored-gse.

---

[1]ENN Science and Technology Development Co., Ltd., Langfang, 065991, China [2]College of Mechanical Engineering, Xi'an University of Science and Technology, Xi'an 710054, China [3]MOE Key Laboratory of Thermo-Fluid Science and Engineering, School of Energy and Power Engineering, Xi'an Jiaotong University, Xi'an 710049, China [4]School of Physics, Dalian University of Technology, Dalian 116024, China. Correspondence to: Tianyuan Liu <liutianyuan@enn.cn>.

*Proceedings of the 43rd International Conference on Machine Learning*, Seoul, South Korea. PMLR 306, 2026. Copyright 2026 by the author(s).

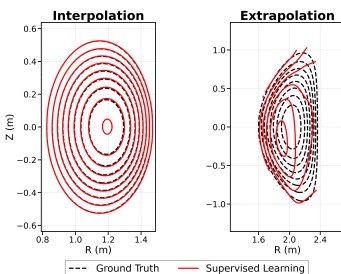

*Figure 1.* Limitation of data-driven surrogates: a data-only model achieves excellent interpolation (left) but produces non-physical flux topology under extrapolation (right).

## 1. Introduction

Fusion energy and artificial intelligence share a symbiotic future: fusion can provide clean power for AI infrastructure, while AI can enable the rapid predictions needed for reactor control. Recent breakthroughs demonstrate AI's transformative potential for fusion: deep reinforcement learning has achieved autonomous magnetic control of tokamak plasmas (Degrave et al., 2022) and active prevention of tearing instabilities (Seo et al., 2024), while neural operators have enabled high-resolution weather forecasting (Kurth et al., 2023) and climate modeling (Kochkov et al., 2024), showcasing their capability for complex physical system emulation. A key computational bottleneck in magnetic confinement fusion is solving the Grad–Shafranov equation (GSE) (Grad & Rubin, 1958; Shafranov, 1966), a strongly nonlinear elliptic PDE that determines axisymmetric tokamak equilibria and underpins equilibrium reconstruction, control, and design.

For scientific simulation, there is a persistent trade-off between accuracy and latency. High-fidelity numerical solvers (Lütjens et al., 1996; Lao et al., 1985) provide precise solutions but are computationally expensive and often unsuitable for real-time or large-scale exploration (Meneghini et al., 2015). Data-driven surrogates (Joung et al., 2020; Zheng et al., 2024) offer fast inference but may violate physical constraints and fail unpredictably when deployed beyond the training distribution—a challenge known as *out-of-distribution (OOD) generalization*. Figure 1 illus-

trates this fundamental limitation on GSE: while a data-only model achieves near-perfect interpolation within the training regime (in-distribution, IID), it collapses to non-physical flux topology under boundary extrapolation.

This OOD failure is not unique to fusion or GSE; it reflects a broader challenge for neural PDE surrogates across scientific domains. Recent stress tests reveal that AI weather models fail to extrapolate from weaker training events to unseen extremes (Sun et al., 2025), while distribution shifts in boundary conditions or source terms can inflate surrogate errors by more than an order of magnitude (Shikhman, 2026), undermining reliability in safety-critical applications. The core question thus extends beyond any single PDE:

*How can neural PDE surrogates generalize reliably under distribution shift, especially for strongly nonlinear problems?*

Several directions have been explored to improve the reliability of neural PDE surrogates under distribution shift. Physics-informed learning methods such as PINNs (Raissi et al., 2019; Karniadakis et al., 2021) incorporate PDE residuals to improve physical consistency, but residual-based optimization can be fragile for nonlinear PDEs, with ill-conditioned differential losses (Krishnapriyan et al., 2021) and unbalanced gradients across composite objectives (Wang et al., 2021). Neural operators (Lu et al., 2021; Li et al., 2021) enable fast inference for parameterized PDE families, and PINO (Li et al., 2024) further combines operator learning with PDE residual losses. However, in strongly nonlinear regimes, physics-only training can still converge to incorrect solution branches, while data-only operators have no mechanism to enforce physical consistency outside the training regime.

Recent advances provide complementary tools. Architectures such as Transolver (Wu et al., 2024) and ONO (Xiao et al., 2024) improve representation and operator generalization, while generative approaches such as flow matching (Lipman et al., 2023; Kuzhamuratov et al., 2026) offer another route to field prediction. In parallel, OOD generalization has become a central concern for neural PDE solvers and scientific ML deployment (Sun et al., 2025; Shikhman, 2026; Wei et al., 2026); physics-aligned representation learning (Wei et al., 2026) and physics-informed fine-tuning (Feng et al., 2026) offer promising routes. Yet fine-tuning is difficult to reconcile with sub-millisecond tokamak-control latency, motivating zero-shot robustness rather than test-time adaptation.

For GSE specifically, prior neural solvers mainly emphasize IID accuracy, simplified or linear profiles, or a single problem setting (Jang et al., 2024; Rizqan et al., 2025; Zhou & Zhu, 2025; Wang et al., 2024). Table 1 summarizes these works along the axes most relevant to deployment relia-

*Table 1.* Comparison of recent neural network-based GS solvers.

| Study | Nonlin. profiles | Boundary setting | OOD eval. | Paradigm comp. | Arch. comp. |
|---|---|---|---|---|---|
| Jang et al. (2024) | × | Fixed | ✓ | × | × |
| Rizqan et al. (2025) | × | Fixed | ✓ | × | ✓ |
| Zhou & Zhu (2025) | × | Fixed | × | × | × |
| Wang et al. (2024) | ✓ | Free | × | × | × |
| **Ours** | ✓ | Fixed | ✓ | ✓ | ✓ |

bility: nonlinear profiles, boundary setting, extrapolation analysis, learning-paradigm comparison, and architecture comparison. Existing studies do not systematically connect architecture choice, training paradigm, and OOD failure mechanism under strongly nonlinear multi-parameter shifts. We therefore study physics-anchored operator learning as a deployment-oriented zero-shot route: data anchors provide calibration toward the physically relevant solution branch, while PDE constraints regularize the field away from labeled points.

This work advances physics-informed operator learning for tokamak equilibria through the following contributions on the Grad–Shafranov equation:

1. Training-paradigm analysis for strongly nonlinear GSE: We compare Data-only, Physics-only, and Physics-anchored operator learning under IID and multi-parameter OOD settings. The comparison reveals complementary failure modes: data-only surrogates can develop severe OOD tails, whereas physics-only training may converge to incorrect solution branches.

2. Physics-motivated operator architecture: Guided by the Green-function view of the fixed-boundary GSE, we instantiate TKNO with a Galerkin Transformer trunk for global elliptic mixing and a KAN regressor for adaptive nonlinear response. Controlled comparisons validate TKNO as the strongest backbone in our setting, reaching $0.25\%$ mean and $0.42\%$ max IID $L_2(\psi)$ error.

3. Physics-anchored tail-risk mitigation: We show that combining data anchors with PDE constraints reduces severe-error tails on shape-driven and joint shifts, lowering joint-OOD $p_{99}$ error from $53.7\%$ to $44.1\%$. Diagnostics suggest two complementary effects: data anchors reduce branch-scale mismatch and PDE residuals suppress high-frequency error amplification.

4. Operational tokamak-equilibrium comparison: On EXL-50U discharge inputs, the model agrees closely with the device's operational equilibrium solver (RMSE $1.27\%$, $R^2 = 99.78\%$, axis error 5.8 mm) while providing millisecond-level inference. It has been deployed to provide real-time reference for experimental design.

## 2. Problem Formulation

### 2.1. The Grad–Shafranov Equation (GSE) as a Parameterized PDE

The Grad–Shafranov equation (GSE) (Grad & Rubin, 1958; Shafranov, 1966) describes axisymmetric tokamak equilibria. It arises from the ideal MHD force balance $\nabla p = \mathbf{J} \times \mathbf{B}$: under axisymmetry, this 3D vector equation reduces to a 2D scalar PDE for the poloidal flux $\psi(R, Z)$ in cylindrical coordinates $(R, \phi, Z)$ (Goedbloed et al., 2010; Freidberg, 2014):

$$\Delta^* \psi = -\mu_0 R J_\phi(\psi), \tag{1}$$

where the Grad–Shafranov operator $\Delta^* = R\partial_R(R^{-1}\partial_R) + \partial_Z^2$, and the toroidal current density $J_\phi(\psi)$ is a nonlinear source term:

$$J_\phi(\psi) = R\frac{dp(\psi)}{d\psi} + \frac{F(\psi)}{\mu_0 R}\frac{dF(\psi)}{d\psi}, \tag{2}$$

determined by pressure $p(\psi)$ and poloidal current $F(\psi)$ profiles. The nonlinearity arises from realistic profile parameterizations (e.g., the GAQ model (McClain & Brown, 1977); see Appendix A.2) where $p(\psi)$ and $F(\psi)$ depend nonlinearly on $\psi$—the solution itself appears in the source term, creating a self-consistent nonlinear relationship that poses optimization challenges for physics-informed learning.

**Input/output and boundary condition.** We consider a fixed-boundary setting where the neural operator predicts the flux field $\psi(R, Z)$ on a discretized computational domain, with the plasma boundary determined by shape parameters and the Dirichlet condition $\psi = 0$ imposed on that boundary. Within this geometry and GAQ source-profile family, each fixed-boundary problem is specified by a 10-dimensional vector covering boundary geometry, global operating quantities, and nonlinear source-profile shape:

$$\xi = (\underbrace{R_0, a, \kappa, \delta}_{\text{shape}}, \underbrace{I_p, \beta_0}_{\text{global}}, \underbrace{n_p, m_p, n_f, m_f}_{\text{profile}}),$$

where the first four entries define the D-shaped boundary, the next two specify the plasma-current scale and pressure/current partition, and the last four control source-profile peakedness. Detailed formulas for the boundary geometry and GAQ source profiles, including the roles of $I_p$, $\beta_0$, and the profile exponents, are given in Appendices A.3 and A.2.

### 2.2. Operator Learning under Multi-Parameter Distribution Shift

We view GSE solving as learning a solution operator $G^\star : \Xi \to \mathcal{U}$ mapping the parameter vector $\xi$ to the solution field $\psi$. Given training pairs $\{(\xi_i, \psi_i)\}_{i=1}^N$, we learn $\hat{G}_\theta \approx G^\star$.

**Distribution shift in fusion operations.** We train on $P_{\text{train}}(\xi)$ and evaluate on $P_{\text{test}}(\xi) \neq P_{\text{train}}(\xi)$. Training data typically come from simulations that cover a limited parameter regime, while deployment scenarios—such as new configurations, higher-current operation, or profile exploration—often require equilibria outside this regime. Because shape, global, and source-profile parameters span physically distinct mechanisms, we evaluate OOD robustness with two complementary views: a *joint* pool in which several parameter groups may go out of range simultaneously (reflecting deployment-style usage), and a *group-isolated* pool in which only one parameter group is allowed out of range while the other two stay strictly in range (isolating the causal contribution of each group). The exact construction is described in Section 4.1.

## 3. Methodology

### 3.1. Overall Framework

Our framework addresses the identified sim-to-real challenge through three complementary components: (i) *operator learning* enables fast inference across parameterized equilibria without retraining; (ii) *physics constraints* enforce differential structure to improve consistency beyond training data; and (iii) *data anchors* provide limited observational calibration for the strongly nonlinear PDE loss.

We follow a three-stage pipeline (Figure 2): (i) generate a dataset using a numerical solver, yielding pairs $\{(\xi_i, \psi_i)\}_{i=1}^N$; (ii) train a physics-informed neural operator $\hat{G}_\theta$ to approximate $G^\star$; (iii) evaluate accuracy, physics consistency, and OOD robustness across shape, global, and source-profile parameter shifts.

We minimize an empirical composite objective:

$$\theta^\star = \arg\min_\theta \frac{1}{N}\sum_{i=1}^N \mathcal{L}_{\text{total}}(\hat{G}_\theta; \xi_i, \psi_i). \tag{3}$$

By adjusting loss weights and supervision availability, we study three training paradigms: **Data-only** (full-field supervised fitting), **Physics-only** (PDE/constraint losses without labels), and **Physics-anchored** (data anchors + physics constraints). Sparse-anchor variants of the Physics-anchored setting are used to study the accuracy–residual trade-off under limited labeled information.

### 3.2. Neural Operator Architecture

Inspired by DeepONet (Lu et al., 2021), we adopt a Branch–Trunk formulation: a **BranchNet** $B$ encodes the input $\xi$ into a latent vector, a **TrunkNet** $T$ encodes coordinates $x = (R, Z)$, and a regressor $R$ maps fused features to the scalar field $\psi$:

$$\hat{\psi}(x) \approx \hat{G}_\theta(\xi)(x) = R\big(B(\xi) \odot T(x)\big). \tag{4}$$

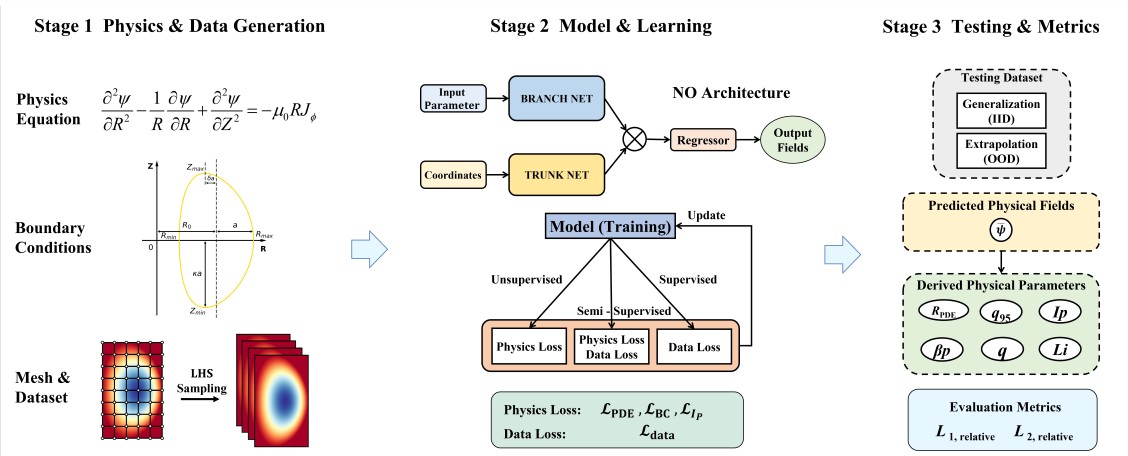

*Figure 2.* Overall framework: data generation, operator learning, and evaluation.

For a fixed-boundary elliptic GSE, Green-function and boundary-integral formulations of the Grad–Shafranov operator (Itagaki & Fukunaga, 2006) suggest the schematic integral form

$$\psi(x) = \int_{\Omega(\xi)} G_\xi(x, x')\, S(x', \psi(x'); \xi)\, dx', \qquad (5)$$

where $G_\xi$ encodes geometry-dependent global elliptic coupling and $S$ denotes the nonlinear source induced by pressure and current profiles. This integral view suggests two modeling requirements. First, the trunk should support nonlocal spatial mixing, because the flux at any point is coupled to the source over the whole plasma domain. Self-attention provides full-field token aggregation, loosely matching the nonlocal mixing suggested by the integral-operator view. In the Galerkin attention used here, each head can be written schematically as

$$Y^{(h)} = Q^{(h)} C^{(h)}, \qquad C^{(h)} = \frac{(K^{(h)})^\top V^{(h)}}{N}, \quad (6)$$

where $C^{(h)}$ is a global feature-coupling matrix aggregated over all spatial tokens. Second, the regressor should represent a smooth but highly nonlinear source-response map. Recent spectral analysis of fixed-boundary GS equilibria shows compact solution structure: using an MXH–Chebyshev basis, Xie & Li (2026) report $10^{-2}$–$10^{-3}$ relative accuracy with only 13–20 parameters in representative fixed-boundary settings. This motivates a KAN regressor with adaptive spline nonlinearities. We do not claim that the learned attention matrix equals the Green's kernel $G_\xi(x, x')$, nor that KAN explicitly recovers the MXH–Chebyshev basis. Rather, these views motivate the TKNO design: a Transformer trunk for global elliptic mixing and a KAN regressor for adaptive nonlinear response. The empirical evidence for this choice comes from the controlled architecture comparisons in Section 4.2, while the mechanistic

claims about the training paradigm are supported separately by quantitative probes in §4.4.1 and Appendix D.3.

We use an MLP as BranchNet, and benchmark multiple Trunk/Regressor choices (MLP, KAN (Liu et al., 2025), FNO (Li et al., 2021), CNN, Transformer (Vaswani et al., 2017; Cao, 2021)) under a unified setting. We denote models as $\{\texttt{trunk}\}\text{--}\{\texttt{regressor}\}$ with abbreviations $M/K/F/C/T$; for example, **T–K** denotes a Transformer trunk with a KAN regressor (which we call **TKNO** for convenience). Since the BranchNet handles only low-dimensional parameter encoding ($\xi \in \mathbb{R}^{10}$), a simple MLP suffices; the TrunkNet and Regressor are responsible for learning the complex spatial structure and are therefore the focus of our benchmark. The comparative effectiveness of different instantiations is evaluated in Section 4.2; architecture diagrams, data-flow descriptions, and inductive-bias analysis for each variant are provided in Appendix B.1.

### 3.3. Physics-Anchored Learning

We train with a composite loss combining data fidelity and physics constraints. The physics loss enforces the discretized PDE residual and boundary/global constraints (e.g., plasma current):

$$\mathcal{L}_{\text{total}} = \omega_{\text{data}}\mathcal{L}_{\text{data}} + \omega_{\text{phys}}\mathcal{L}_{\text{phys}}, \qquad (7)$$

$$\mathcal{L}_{\text{phys}} = \omega_{\text{PDE}}\mathcal{L}_{\text{PDE}} + \omega_{\text{BC}}\mathcal{L}_{\text{BC}} + \omega_{I_p}\mathcal{L}_{I_p}. \qquad (8)$$

For architectures where automatic differentiation is inconvenient (e.g., Transformer, CNN, FNO with FFT operations), we compute the PDE residual using a finite difference (FD) discretization that matches the ground-truth solver; a comparison of FD vs. AD is provided in Appendix B.2. By selecting loss weights, we explore:

- **Data-only**: $\omega_{\text{data}} > 0$, $\omega_{\text{phys}} = 0$, using full-field supervision.

- **Physics-only**: $\omega_{\text{data}} = 0$, $\omega_{\text{phys}} > 0$, using only PDE and physical constraints.

- **Physics-anchored**: $\omega_{\text{data}} > 0$, $\omega_{\text{phys}} > 0$, using data anchors together with PDE/BC/$I_p$ constraints. Sparse-anchor variants use only a subset of labeled points per sample for $\mathcal{L}_{\text{data}}$.

**Anchor mechanism.** For strongly nonlinear PDEs, a small residual is necessary but not sufficient: multiple solution branches can satisfy the residual approximately, and Physics-only optimization may converge to a non-physical branch. Data anchors add observational information that helps select the physically relevant branch, while the PDE/BC/$I_p$ constraints maintain physical consistency away from labeled points. Dense supervision is not automatically better because excessive data gradients can conflict with residual minimization and push the model toward data-dominated interpolation. In Section 4.4.1, we test this interpretation with branch-identification and spectral-amplification diagnostics.

**Training protocol.** All models are trained with the Adam optimizer and a multi-step learning-rate schedule. Loss weights are tuned per paradigm; detailed hyperparameter settings are provided in Appendix B.3.

# 4. Evaluation

## 4.1. Experimental Setup

**Datasets.** We generate equilibrium solutions using a high-fidelity finite difference numerical solver with successive over-relaxation (FDM+SOR). We use two dataset settings. The pilot 4D setting varies only the boundary-shape parameters $(R_0, a, \kappa, \delta)$, with global and profile quantities fixed, for cost-efficient architecture validation and training-paradigm diagnostics. The final robustness study uses the full 10D ranges in Table 2. In Table 3, IID splits are reported as train/validation/test; the held-out test portion is used as the 1k IID reference for the final OOD analysis. Following the joint and group-isolated definitions in Section 2.2, the OOD evaluation uses a joint pool of 10k samples and an isolated pool of 3k samples, balanced as 1k shape, 1k global, and 1k profile cases.

**Metrics.** We report relative errors on the flux field $\psi$ (shown as % in tables) and physics consistency via PDE-residual statistics MAE(PDE) / MSE(PDE) (in original units; computed by finite differences on the predicted field). Unless stated otherwise, we highlight the best value in each column in bold and the second-best underlined; values are rounded for display. Derived physical quantities ($I_p, q_{95}, \beta_p, l_i$) are compared in Appendix D. Formal metric definitions are provided in Appendix C.

*Table 2.* Full 10D parameter ranges. The IID range is used for training and in-range testing; the OOD range defines the wider sampling domain for joint and group-isolated extrapolation.

| Group | Parameters | IID range | OOD range |
|---|---|---|---|
| Shape | $R_0$ | $[1.0, 1.5]$ | $[0.5, 2.0]$ |
| | $a$ | $[0.3, 0.5]$ | $[0.1, 0.7]$ |
| | $\kappa$ | $[1.0, 2.5]$ | $[0.5, 3.0]$ |
| | $\delta$ | $[-0.5, 0.5]$ | $[-0.9, 0.9]$ |
| Global | $I_p$ | $[3, 7] \times 10^5$ | $[1, 10] \times 10^5$ |
| | $\beta_0$ | $[0.5, 0.8]$ | $[0.3, 1.0]$ |
| Profile | $n_p, m_p, n_f, m_f$ | $\{1, 2\}$ | $\{1, 2, 3, 4\}$ |

*Table 3.* Dataset usage and splits. All samples use the same $101 \times 161$ grid.

| Stage | Input dim. | IID split | OOD eval. |
|---|---|---|---|
| Pilot | 4D (shape) | 6k/2k/2k | – |
| Final | 10D (full) | 9k/1k/1k | Joint 10k isolated 3k |

## 4.2. Hypothesis-Driven Architecture Validation

The methodology suggests a specific architectural hypothesis rather than an unconstrained model search: fixed-boundary GSE surrogates should combine global elliptic mixing with an adaptive nonlinear source-response map. We therefore first use Data-only training to isolate representation capacity before adding physics losses. This supervised-first protocol addresses a narrow question: under the same data and loss, which backbone best matches the operator structure of the GSE?

Figure 3 and Table 4 serve complementary roles. The figure shows the broader architecture sweep and representative flux/error maps, while the table reports the close numerical comparisons used for model selection. The pointwise MLP/KAN comparison probes nonlinear source-response representation; the fixed-KAN trunk comparison isolates spatial mixing; and ONO, Transolver, and Flow Matching provide recent cross-family references. Full trunk–regressor variants are retained in Appendix D.

**Architecture-physics alignment.** The main pattern is compact: KAN sharply reduces flux error relative to a plain coordinate MLP, and with KAN fixed as the regressor, the Transformer trunk is the only variant that is simultaneously strongest in mean flux accuracy, worst-case flux accuracy, and finite-difference residual. CNN attains competitive pointwise error but a much larger PDE residual, indicating that local interpolation accuracy alone does not recover the differential structure.

**Comparison with recent baselines.** TKNO remains competitive against ONO, Transolver, and Flow Matching in the same pilot setting. Flow Matching is a useful generative reference but requires iterative sampling (32 Heun steps,

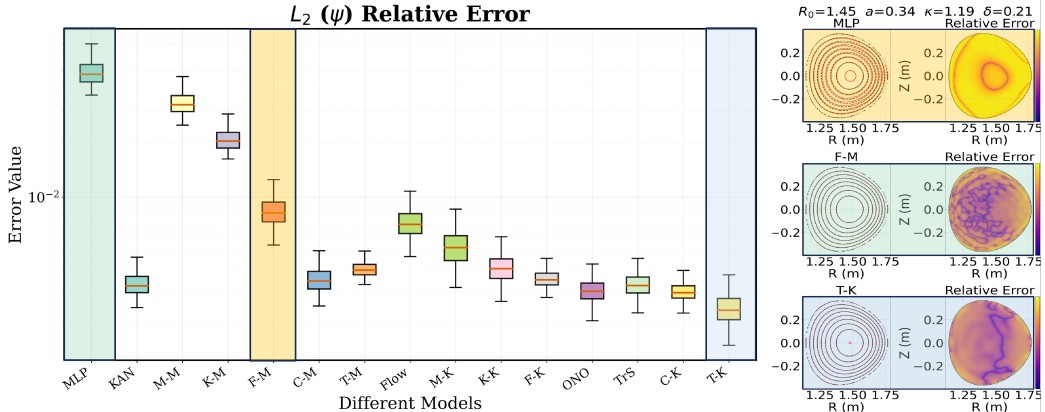

*Figure 3.* Data-only architecture sweep on the pilot IID test set. Left: distribution of $L_2(\psi)$ relative error across model variants. Right: representative flux predictions and relative-error maps. Table 4 reports the focused numerical comparison used to select the TKNO backbone.

*Table 4.* Focused architecture comparison under Data-only training. Flux errors are on the pilot IID test set and reported in %; PDE residuals are finite-difference MAE statistics in original units. Flow Matching reports the best U-Net variant among five tested backbones and does not directly provide a PDE-residual statistic.

| Model | Family / role | Params (M) | Mean $L_2(\psi)$ | Max $L_2(\psi)$ | MAE(PDE) |
|---|---|---|---|---|---|
| *Pointwise nonlinear-response baselines* | | | | | |
| MLP | coordinate MLP | 0.13 | 4.93 | 8.34 | 1.51 |
| KAN | coordinate KAN | 0.34 | 0.34 | 0.66 | 2.36 |
| *Fixed-KAN trunk variants* | | | | | |
| M–K | MLP trunk + KAN | 0.24 | 0.54 | 0.92 | 2.46 |
| F–K | FNO trunk + KAN | 0.30 | 0.36 | 0.62 | 3.16 |
| C–K | CNN trunk + KAN | 0.47 | 0.30 | 0.45 | 5.23 |
| T–K (TKNO) | Transformer trunk + KAN | 0.29 | **0.25** | **0.42** | **1.23** |
| *Recent cross-family baselines* | | | | | |
| Flow Matching | generative transport (best U-Net) | 0.91 | 0.74 | 1.40 | – |
| Transolver | physics-attention operator | 0.31 | 0.33 | 0.51 | 2.46 |
| ONO | orthogonal-attention operator | 0.29 | 0.31 | 0.48 | 1.61 |

approximately 100 ms in our implementation), which is less suitable for sub-millisecond equilibrium use. We therefore select TKNO not because any module is novel in isolation, but because this combination best matches the GSE structure under controlled comparison.

These results fix **T–K** as the primary backbone for the remaining experiments. Architecture choice alone, however, does not determine how the surrogate behaves away from labeled data. We therefore next keep the backbone fixed and vary the training paradigm to test the roles of residual constraints and data anchors.

### 4.3. Physics-Informed Training Diagnostics

Before the final 10D OOD evaluation, we use the pilot 4D setting to examine the training paradigms. We focus on the selected TKNO backbone and include F–K as a representative Fourier neural-operator baseline. This section asks two questions. First, if the model is trained only by

the GSE residual, does it recover the correct equilibrium field? Second, after adding sparse data anchors, how much supervision is needed to improve accuracy while still keeping the PDE residual small? Unless noted otherwise, PDE residuals are computed using finite differences to match the discretization of the ground-truth solver.

#### 4.3.1. PHYSICS-ONLY TRAINING AS A RESIDUAL-ONLY CONTROL

We first train PINO models using physics-only losses ($\omega_{\text{data}} = 0$). This residual-only setting tests the tempting but incomplete hypothesis that enforcing the GSE residual is sufficient. In addition to TKNO and FNO, we attempted physics-only training on other architectures (MLP/KAN/CNN trunks) with hyperparameter tuning over PDE-loss weights and learning-rate schedules. Representative training dynamics, tuning ranges, and failure cases are deferred to Appendix D.2.

*Table 5.* Physics-only training on IID ($\psi$ error in %; MSE(PDE) in original units).

| Model | Train Dataset | | Test Dataset | | MSE (PDE) |
|---|---|---|---|---|---|
| | $L_1(\psi)$ | $L_2(\psi)$ | $L_1(\psi)$ | $L_2(\psi)$ | (Test) |
| MLP | 10.10 | 9.50 | 11.40 | 10.70 | 0.835 |
| KAN | 6.81 | 6.10 | 6.86 | 6.13 | 0.429 |
| F–K | 2.68 | 2.48 | 1.86 | 1.73 | 0.080 |
| C–K | 5.59 | 5.09 | 5.83 | 5.30 | 0.442 |
| T–K | **0.58** | **0.62** | **0.59** | **0.62** | **0.003** |

*Table 6.* Physics-anchored IID results at representative sparse-point budgets ($\psi$ error in %; MSE(PDE) in original units).

| Model | Points | Train Dataset | | Test Dataset | | MSE (PDE) |
|---|---|---|---|---|---|---|
| | | $L_1(\psi)$ | $L_2(\psi)$ | $L_1(\psi)$ | $L_2(\psi)$ | (Test) |
| T–K | 10 | 0.70 | 0.75 | 0.70 | 0.75 | **0.007** |
| | 100 | 0.50 | 0.59 | 0.51 | 0.59 | 0.011 |
| | 1000 | **0.42** | **0.52** | **0.41** | **0.52** | 0.023 |
| F–K | 10 | 1.66 | 1.60 | 1.66 | 1.60 | 0.192 |
| | 100 | 0.83 | 1.02 | 0.83 | 1.02 | 0.122 |
| | 1000 | 0.62 | 0.81 | 0.63 | 0.82 | 0.118 |

**Physics-only results.** Table 5 shows that physics-only training can drive the PDE residual down, especially for TKNO and FNO, but the resulting flux-field accuracy remains worse than Data-only training in Table 4: the best physics-only TKNO reaches $0.621\%$ test $L_2(\psi)$, about $2.5\times$ the Data-only TKNO error. MLP/KAN/CNN variants are harder to optimize under comparable tuning effort, and Appendix D.2 shows that physics-only training is sensitive to residual-weight choices. Residual minimization alone is therefore difficult to optimize and may converge to low-residual fields that are not the desired equilibrium. This motivates adding data anchors to guide the optimization toward the relevant solution branch.

4.3.2. DATA-ANCHOR ROLE AND BUDGET DIAGNOSTIC

Next, we introduce data anchors together with physics constraints ($\omega_{\text{data}} > 0$, $\omega_{\text{phys}} > 0$). The purpose of the sparse-point sweep is diagnostic: it tests how limited anchor information changes the accuracy–residual balance, with F–K included as a classical neural-operator reference under the same anchored objective. The final 10D OOD study uses the calibrated Physics-anchored paradigm; the sparse-anchor sweep should be read as evidence for the role of anchors, not as a claim that one point budget is universally optimal.

**Sparse-anchor trade-off.** Table 6 shows that sparse anchors substantially improve solution accuracy while keeping the PDE residual far below the Data-only residual scale in Table 4. For **TKNO**, increasing the budget from 10 to 100 points captures most of the accuracy gain; increasing to 1000 points does not monotonically improve the accuracy–residual trade-off. This supports the branch-selection interpretation without implying a universally optimal point budget. Under the same budgets, **TKNO consistently outperforms FNO**, reinforcing the architecture choice under physics-anchored training.

Together, these diagnostics establish the two ingredients needed for the final robustness test: a GSE-aligned TKNO backbone, and a Physics-anchored objective in which data signals select the branch while PDE residuals regularize the field. We next evaluate whether this combination reduces OOD collapse under full 10D multi-parameter distribution

shift.

### 4.4. OOD Robustness under Multi-Parameter Distribution Shift

We next test the selected TKNO backbone and three training paradigms in the full 10D setting. The IID reference is a dedicated test split sampled from the in-distribution ranges and not used for training or validation. OOD evaluation follows the group-isolated and joint-pool protocol defined in Section 2.2, allowing us to separate single-group effects from deployment-style multi-parameter shifts.

**Bulk accuracy vs. tail risk.** As shown in Table 7, Data-only has the lowest mean error on IID and on three OOD subsets, so it remains the stronger bulk predictor. Physics-anchored is slightly worse in mean error on Global, Profile, and Joint shifts, but the gap is modest compared with the OOD growth and tail behavior. Because Physics-anchored starts from a higher IID error, its OOD/IID growth is consistently smaller than Data-only; more importantly, its worst-tail errors are lower on shape-driven and joint shifts, reducing $p_{99}$ from $52.3\%$ to $33.2\%$ on isolated Shape and from $53.7\%$ to $44.1\%$ on Joint Any-OOD. Physics-only shows small growth ratios only because its IID baseline is already poor, and its absolute OOD errors remain the largest.

**Scope of the gain.** The evidence should therefore be read as a trade-off rather than uniform dominance: Data-only gives better average accuracy, while Physics-anchored gives more graceful degradation and lower severe-error tails on the shifts most tied to boundary geometry. Isolated Profile is the clearest remaining gap, where Data-only dominates at every percentile. Additional aggregate OOD views by parameter, shift pattern, and shift multiplicity are provided in Appendix D.3.

**Qualitative cases.** Figure 4 visualizes challenging large-extrapolation cases rather than average OOD behavior. The three cases are drawn from joint-OOD settings with multiple out-of-range input coordinates, covering a large-radius/profile-shift case, a compact low-elongation/global-shift case, and a high-elongation case. They are intended

*Table 7.* $L_2(\psi)$ relative error (%) across IID and OOD subsets for the three training paradigms. "Growth" is the ratio of mean OOD error to the corresponding IID baseline; $p_{95}$ and $p_{99}$ summarize tail risk.

| Subset | Paradigm | Mean ± std (%) | Growth (×) | $p_{95}$ (%) | $p_{99}$ (%) |
|---|---|---|---|---|---|
| IID | Data-only | **0.24±0.06** | — | **0.34** | **0.42** |
| | Physics-anchored | 0.59±0.10 | — | 0.78 | 0.91 |
| | Physics-only | 6.82±1.41 | — | 9.61 | 10.91 |
| Iso. Shape | Data-only | 7.11±10.62 | 29.9 | 29.04 | 52.25 |
| | Physics-anchored | **6.83±7.49** | **11.6** | **23.10** | **33.16** |
| | Physics-only | 27.10±30.42 | 4.0 | 83.29 | 160.16 |
| Iso. Global | Data-only | **2.42±5.19** | 10.2 | **14.29** | 27.92 |
| | Physics-anchored | 2.76±4.82 | **4.7** | 14.30 | **25.77** |
| | Physics-only | 9.63±8.16 | 1.4 | 28.60 | 48.11 |
| Iso. Profile | Data-only | **5.80±4.43** | 24.4 | **14.09** | **22.47** |
| | Physics-anchored | 7.69±5.76 | **13.1** | 18.46 | 23.95 |
| | Physics-only | 21.35±27.04 | 3.1 | 85.56 | 130.47 |
| Joint Any-OOD | Data-only | **10.31±10.80** | 43.3 | 31.62 | 53.66 |
| | Physics-anchored | 11.44±11.25 | **19.5** | **28.60** | **44.12** |
| | Physics-only | 42.77±64.36 | 6.3 | 127.41 | 281.52 |

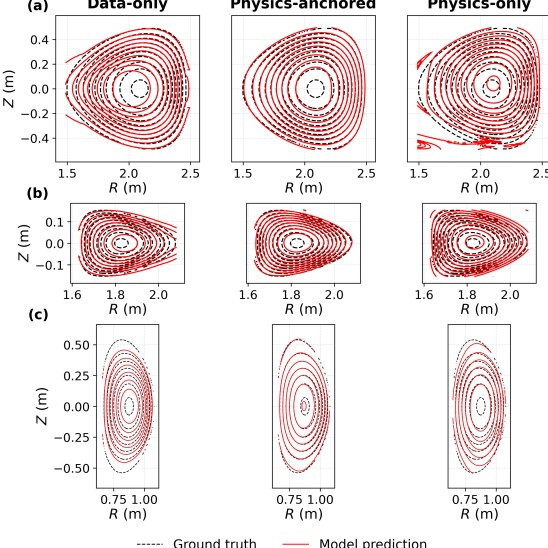

*Figure 4.* Challenging large-extrapolation joint-OOD cases. Red solid contours are model predictions and black dashed contours are ground truth. Case inputs are listed in Appendix D.3.

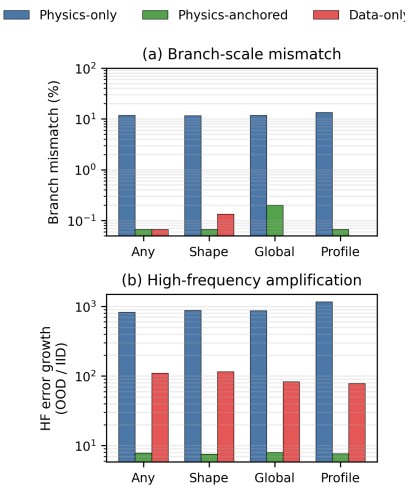

*Figure 5.* Mechanism diagnostics on joint-pool OOD slices. (a) Branch-scale mismatch. (b) High-frequency error growth.

to illustrate the tail-risk pattern in Table 7: Data-only can be the better bulk predictor, but in severe extrapolation it may produce visible contour distortions, whereas Physics-anchored better preserves the nested flux-surface structure.

### 4.4.1. MECHANISM DIAGNOSTICS

The OOD results show where the models degrade; we next use two diagnostic views to interpret how they fail. These diagnostics are not causal ablations. Instead, they summarize two observable error patterns: branch-scale mismatch

and high-frequency error amplification. Full definitions and tables are provided in Appendix D.3.

**Branch-scale mismatch.** We first check whether the predicted flux has the correct sign and a comparable magnetic-axis magnitude. For case $i$, let $\mathcal{P}_i$ indicate that the prediction contains a plasma region with the correct sign, and define the axis-magnitude ratio and failure rate as

$$r_i = \frac{|\min_x \hat{\psi}_i(x)|}{|\min_x \psi_i(x)|},$$

$$F_{\text{branch}} = 1 - \frac{1}{N}\sum_{i=1}^{N}\mathbf{1}[\mathcal{P}_i \wedge r_i \in [1/3, 3]]. \tag{9}$$

Figure 5(a) plots $100F_{\text{branch}}$. This coarse criterion is not

a replacement for flux-field error; it is a sanity check for wrong-sign or wrong-scale equilibria. Physics-only violates the criterion on $\approx 12\%$ of OOD cases, whereas adding either sparse anchors or dense labels reduces the rate below $0.2\%$. This supports the interpretation that data supervision helps steer optimization toward the relevant equilibrium branch.

**High-frequency amplification.** Among models that mostly pass the branch-scale check, OOD errors can still appear as oscillatory or high-frequency artifacts. Following standard spectral-error diagnostics for neural PDE surrogates (Shikhman, 2026), we compute the high-frequency error energy and its OOD/IID growth for the error field $e_i = \hat{\psi}_i - \psi_i$:

$$E_{\mathrm{HF}}(\mathcal{D}) = \frac{1}{|\mathcal{D}|} \sum_{i \in \mathcal{D}} \sum_{\|k\|/k_{\max} > 0.35} |\mathcal{F}[e_i](k)|^2,$$

$$G_{\mathrm{HF}}(\mathcal{D}_{\mathrm{OOD}}) = \frac{E_{\mathrm{HF}}(\mathcal{D}_{\mathrm{OOD}})}{E_{\mathrm{HF}}(\mathcal{D}_{\mathrm{IID}})}. \tag{10}$$

Figure 5(b) plots $G_{\mathrm{HF}}$ for each OOD slice. Physics-anchored limits high-frequency amplification to $7.5$–$8.1\times$, compared with $78$–$116\times$ for Data-only and $830$–$1170\times$ for Physics-only. Thus, after data anchors help select the relevant branch, the residual term appears to act mainly as a field regularizer that suppresses OOD-induced high-frequency error.

### 4.5. Real Tokamak Operational Equilibrium Comparison

Finally, we test whether the surrogate transfers to experimental operating scenarios by evaluating it on EXL-50U discharge inputs. Because the internal poloidal-flux field is not directly measurable, Figure 6 compares our predictions against the standard equilibrium (SE) solver—the device's operational equilibrium design and feedforward waveform generation program used for experimental planning.

Across multiple discharge shots, our model agrees closely with the SE solver inside the LCFS: RMSE $1.27\pm0.44\%$, $R^2 = 99.78\pm0.13\%$, magnetic-axis error $5.8\pm3.9$ mm, LCFS Hausdorff distance $19.3\pm2.1$ mm, and elongation error $|\Delta\kappa| = 0.009\pm0.007$. The model reduces inference from seconds to milliseconds, providing a real-time equilibrium reference for experimental design and operation. Appendix C.4 defines the reported operational-comparison metrics.

## 5. Conclusion

This work addresses robust and fast operator learning for the strongly nonlinear Grad–Shafranov equation, a central component of tokamak equilibrium prediction. We develop a physics-anchored framework that combines sparse data anchors with PDE-residual constraints, and instantiate it with a physics-motivated TKNO backbone designed to cap-

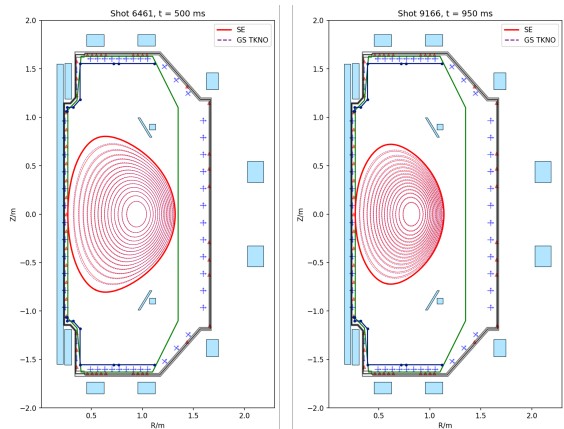

*Figure 6.* Operational equilibrium comparison on EXL-50U discharge inputs: TKNO predictions (dashed) vs standard equilibrium solver (SE, solid) on two shots. Both shots show close agreement in flux-surface topology.

ture global elliptic coupling and nonlinear source response. The study focuses on fixed-boundary equilibria under multi-parameter distribution shifts, where reliable zero-shot extrapolation is essential for deployment-oriented fusion workflows.

Our evaluation connects architecture choice, training paradigm, and OOD behavior. Under controlled Data-only comparisons, TKNO provides the strongest backbone in our setting, reaching $0.25\%$ mean and $0.42\%$ max IID $L_2(\psi)$ error. However, the full 10D OOD study shows that low IID error alone is not sufficient: Data-only remains the stronger bulk predictor on average, but its joint-OOD mean grows by $43.3\times$ and its $p_{99}$ error reaches $53.7\%$. Physics-anchored training gives a more graceful degradation profile, reducing joint-OOD growth to $19.5\times$ and $p_{99}$ to $44.1\%$, with the strongest tail reduction on shape-driven shifts ($52.3\% \rightarrow 33.2\%$). Diagnostic probes further suggest a two-part mechanism: data anchors reduce branch-scale mismatch from $\approx 12\%$ to below $0.2\%$, while residual constraints suppress high-frequency error growth from $78$–$116\times$ to $7.5$–$8.1\times$.

Finally, we evaluate the Physics-anchored TKNO on EXL-50U discharge inputs by comparing against the device's operational standard-equilibrium solver, since the internal poloidal-flux field is not directly measurable. The model achieves close agreement inside the LCFS (RMSE $1.27\pm0.44\%$, $R^2 = 99.78\pm0.13\%$, axis error $5.8\pm3.9$ mm) and reduces inference from seconds to milliseconds, providing a practical real-time equilibrium reference for experimental design and operation. These results support physics-anchored operator learning as a practical route toward physically reliable AI surrogates for fusion workflows, while leaving free-boundary equilibria and broader device-to-device generalization as important next steps.

## Acknowledgements

The authors gratefully acknowledge ENN Science and Technology Development Co., Ltd. for scientific research funding and institutional support for this study. We also thank Tao Zhang for assistance with the flow-matching baseline experiments.

## Impact Statement

This paper presents work whose goal is to advance the field of Machine Learning. There are many potential societal consequences of our work, none which we feel must be specifically highlighted here.

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

# Appendix

This appendix provides additional details and experimental results to support the main submission.

- In Appendix C, we define evaluation metrics including flux field accuracy ($L_1$, $L_2$), PDE residual, derived physical quantities ($q_{95}$, $\beta_p$, $l_i$), and operational equilibrium comparison metrics.

## A. Data Generation

The Grad–Shafranov equation follows from the ideal MHD equilibrium condition. Under the assumption of axisymmetry ($\partial/\partial\phi = 0$), the 3D force balance $\nabla p = \mathbf{J} \times \mathbf{B}$ together with Maxwell's equations $\nabla \cdot \mathbf{B} = 0$ and $\nabla \times \mathbf{B} = \mu_0 \mathbf{J}$ reduces to a single 2D elliptic PDE for the poloidal flux $\psi(R, Z)$ (Goedbloed et al., 2010; Freidberg, 2014). Specifically, the magnetic field can be written as $\mathbf{B} = \nabla\psi \times \nabla\phi/R + F(\psi)\nabla\phi$, where $F(\psi) = RB_\phi$ is the poloidal current function. Substituting into the force balance and using axisymmetry yields the GSE. The detailed derivation can be found in standard MHD textbooks (Goedbloed et al., 2010; Freidberg, 2014); here we focus on the computational aspects relevant to data generation.

### A.1. Dataset Settings and Splits

All samples are generated by the same FDM+SOR solver on a $101 \times 161$ grid. The pilot 4D setting varies only the boundary-shape parameters $(R_0, a, \kappa, \delta)$ and fixes the global/profile parameters at nominal in-range values; it uses an IID train/validation/test split of 6k/2k/2k samples for architecture selection and training-paradigm analysis. The final robustness study uses the full 10D parameterization and ranges in Table 2.

For the full 10D IID pool, we draw 11,000 samples uniformly from the IID ranges in Table 2: 9,000 for training, 1,000 for validation, and 1,000 held-out test samples. The held-out test split is reused as the 1,000-sample in-range reference for OOD analysis. The joint OOD pool contains 10,000 samples drawn from the wider OOD ranges, allowing several parameter groups to leave the training range simultaneously. The isolated OOD pool contains 3,000 samples, 1,000 per group; within each subset only one of the shape, global, or profile groups is allowed out of range while the other two remain in range.

**Physical validity filtering.** Candidate samples are screened before being included in the dataset. In particular, we reject geometries with non-positive inner major radius by requiring $R_0 > a$, so that the rectangular computational domain $R \in [R_0 - a, R_0 + a]$ stays in the physical $R > 0$ region. The sampled ranges also enforce positive elongation and $|\delta| < 1$, keeping the Miller boundary parameterization well defined. Samples that do not yield finite, converged FDM–SOR solutions are not exported. Thus, the OOD sets are stress tests over a wider but numerically valid fixed-boundary GAQ family, rather than arbitrary parameter combinations or a claim that every OOD point corresponds to an operational discharge scenario.

## A.2. Source Term Parameterization

The GSE source term $J_\phi = Rp'(\psi) + FF'(\psi)/(\mu_0 R)$ requires specifying pressure and current profiles. Common parameterizations include:

**Solov'ev model** (linear): $p' = $ const, $FF' = $ const. Admits analytic solutions (Cerfon & Freidberg, 2010) but produces unrealistic peaked current profiles for shaped plasmas.

**Polynomial model**: $p'(\bar{\psi}) = \sum_k a_k \bar{\psi}^k$, $FF'(\bar{\psi}) = \sum_k b_k \bar{\psi}^k$. Flexible but requires many parameters; risk of non-physical oscillations.

**GAQ model** (this work) (McClain & Brown, 1977): We adopt a power-law parameterization for both source functions:

$$p'(\psi) = -\beta_0 \frac{\lambda}{R_0}(1 - \bar{\psi}^{n_p})^{m_p}, \tag{11}$$

$$(F^2(\psi))' = -2\lambda\mu_0 R_0(1 - \beta_0)(1 - \bar{\psi}^{n_f})^{m_f}, \tag{12}$$

where $\bar{\psi} = (\psi - \psi_{\text{axis}})/(\psi_{\text{boundary}} - \psi_{\text{axis}}) \in [0, 1]$ is the normalized poloidal flux, and $R_0$ is the major radius of the device. The current profile is characterized by six parameters: $n_p, m_p, n_f, m_f, \lambda$ (or equivalently a target current scale), and $\beta_0$:

- $n_p, m_p, n_f, m_f$: shape exponents controlling the radial profile of pressure and current.

- $\lambda$ / $I_p$: a scaling factor related to the total plasma current.

- $\beta_0$: a scalar quantity related to the poloidal beta $\beta_p$.

Compared to the Solov'ev equilibria commonly used in existing PINNs' studies (Cerfon & Freidberg, 2010), this model, while more challenging to solve due to its nonlinearity, offers a more realistic representation of actual tokamak plasmas. The architecture and training-paradigm benchmarks use a 4D shape-only pilot setting with global/profile quantities fixed at nominal in-range values, while the final OOD study uses the full 10D parameterization with the IID and OOD ranges of Table 2. The OOD pools allow shape, global, and source-profile groups to drift out of the training range either jointly or in group isolation, as specified in Section 4.1. The GAQ family still introduces **profile-family bias**: the learned operator may not generalize to qualitatively different profile shapes (e.g., reversed-shear, H-mode pedestals), which we discuss as a limitation.

## A.3. Boundary Shape Parameterization

In this work, we consider a fixed-boundary scenario with an elongated, "D"-shaped toroidal geometry. The plasma boundary $\Gamma$ is parameterized by an angle $\tau \in [0, 2\pi]$ and four geometric shape parameters (Miller et al., 1998):

$$R(\tau) = R_0 + a\cos(\tau + \arcsin(\delta)\sin(\tau)), \tag{13}$$

$$Z(\tau) = \kappa a \sin(\tau). \tag{14}$$

These four parameters form the shape subset of the input vector $\xi$:

- $R_0$: major radius, defined as $R_0 = (R_{\max} + R_{\min})/2$

- $a$: minor radius, defined as $a = (R_{\max} - R_{\min})/2$

- $\kappa$: elongation, defined as $\kappa = b/a$ where $b = (Z_{\max} - Z_{\min})/2$

- $\delta$: triangularity, defined as $\delta = (R_0 - R_{Z_{\max}})/a$

These radii define the aspect ratio $A = R_0/a$. For a fixed-boundary problem, the boundary condition is defined by setting the poloidal flux to zero on the plasma boundary:

$$\psi(R, Z)\big|_\Gamma = 0. \tag{15}$$

### A.4. Numerical Solver

To solve the GSE numerically, we employ a finite difference method (FDM) coupled with a Successive Over-Relaxation (SOR) iterative scheme (Xie, 2018). The computational domain is a uniform rectangular grid in the $(R, Z)$ plane with boundaries set at $R \in [R_0 - a, R_0 + a]$ and $Z \in [-\kappa a, \kappa a]$, ensuring that the D-shaped plasma boundary is fully contained within the computational domain for all combinations of shape parameters.

The second-order partial differential GS operator $\Delta^*$ is discretized on a 2D $(R, Z)$ grid ($101 \times 161$ points) using a five-point central difference stencil. The partial derivatives with respect to $R$ and $Z$ at each grid point $(R_i, Z_j)$ are approximated using second-order central differences:

$$\frac{\psi_{i+1,j} - 2\psi_{i,j} + \psi_{i-1,j}}{(\delta R)^2} + \frac{\psi_{i,j+1} - 2\psi_{i,j} + \psi_{i,j-1}}{(\delta Z)^2} - \frac{1}{R_i}\frac{\psi_{i+1,j} - \psi_{i-1,j}}{2\delta R} = S_{i,j}, \tag{16}$$

where $\psi_{i,j} = \psi(R_i, Z_j)$, $\delta R$ and $\delta Z$ are the grid spacings, and the source term is given by:

$$S_{i,j} = -\mu_0 R_i^2 p'(\psi_{i,j}) - f(\psi_{i,j})f'(\psi_{i,j}). \tag{17}$$

The resulting system of nonlinear algebraic equations is solved iteratively using the SOR method. For each grid point $(i, j)$ within the plasma boundary, the poloidal flux $\psi_{i,j}$ is updated at each iteration step $n + 1$ based on the values from step $n$:

$$\psi_{i,j}^{n+1} = (1 - \omega)\psi_{i,j}^n + \frac{\omega}{C}\left(\frac{\psi_{i+1,j}^n + \psi_{i-1,j}^{n+1}}{(\delta R)^2} + \frac{\psi_{i,j+1}^n + \psi_{i,j-1}^{n+1}}{(\delta Z)^2} - \frac{\psi_{i+1,j}^n - \psi_{i-1,j}^{n+1}}{2R_i\delta R} - S_{i,j}^n\right), \tag{18}$$

where $C = \frac{2}{(\delta R)^2} + \frac{2}{(\delta Z)^2}$, and $\omega$ is the relaxation factor chosen to optimize convergence rate (we use $\omega = 1.8$). The iteration continues until the relative error of the poloidal flux field between successive iterations falls below a tolerance ($10^{-8}$) or a maximum number of iterations (50,000) is reached.

## B. Implementation Details

### B.1. Neural Operator Instantiation

To construct an efficient surrogate model for the GSE, we propose a modular Neural Operator (NO) framework designed to learn the solution operator $\mathcal{G}$ that maps the 10D shape/global/profile control parameters to the solution function $\psi(\mathbf{x})$. Inspired by DeepONet (Lu et al., 2021), our architecture decouples parameter dependencies and spatial dependencies into two parallel networks: a Branch Net and a Trunk Net. Figure 7 illustrates the overall architecture.

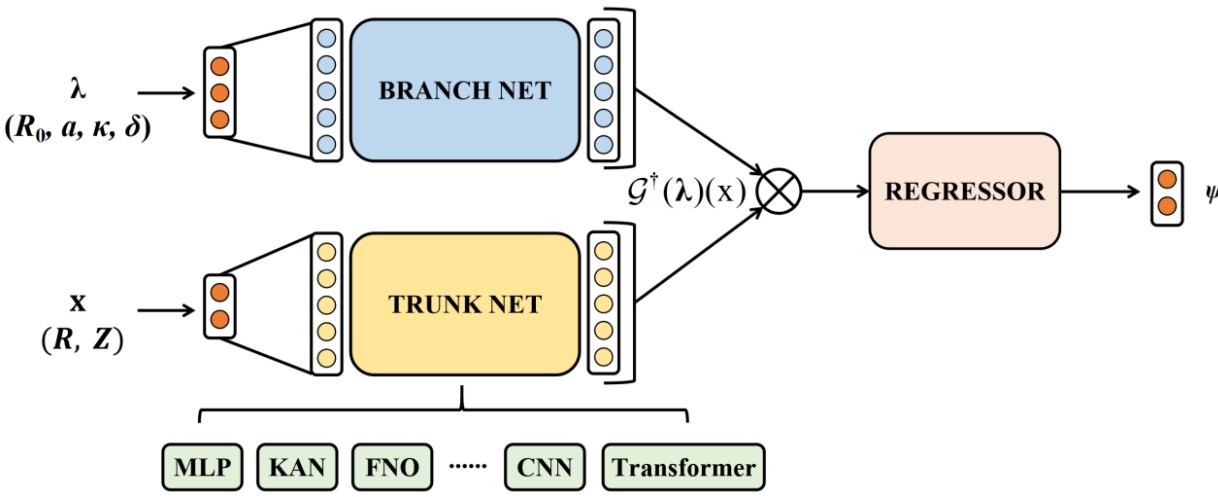

*Figure 7.* Neural operator architecture for GSE solving. The framework consists of three modular components: (a) BranchNet encodes the input parameter vector into a latent feature vector; (b) TrunkNet processes 2D spatial coordinates $(R, Z)$ to produce position-dependent basis functions; (c) the Regressor combines fused features to output the scalar flux field $\hat{\psi}(R, Z)$. The dashed boxes indicate interchangeable architectural variants (MLP, KAN, CNN, FNO, Transformer) benchmarked in this study.

The overall inference process can be mathematically formalized as:

$$\mathcal{G}^{\dagger}(\xi)(\mathbf{x}) \approx \mathcal{R}\big(\mathcal{B}(\xi) \otimes \mathcal{T}(\mathbf{x})\big), \tag{19}$$

where $\mathcal{B}$ is the BranchNet, $\mathcal{T}$ is the TrunkNet, $\otimes$ denotes feature fusion, and $\mathcal{R}$ is the Regressor. This design allows for flexible handling of the mapping between finite-dimensional parameter spaces and infinite-dimensional function spaces.

**Why this three-stage design?**

- **BranchNet** $\mathcal{B}$: Encodes the low-dimensional input parameters ($d = 10$) into a latent feature vector $\mathbf{b} \in \mathbb{R}^p$. Since this input space is low-dimensional, we employ a standard MLP-style parameter encoder for all experiments.

- **TrunkNet** $\mathcal{T}$: Encodes the spatial coordinate grid $\mathbf{x} = (R, Z) \in \mathbb{R}^{H \times W \times 2}$ into a high-dimensional basis function space $\mathbf{T} \in \mathbb{R}^{H \times W \times p}$. The trunk net must process 2D spatial coordinates and capture the complex, nonlinear spatial relationships of the magnetic flux field. Given this critical role, we employ this module as the testbed for benchmarking different deep learning architectures.

- **Regressor** $\mathcal{R}$: Projects the fused features $\mathbf{F} = \mathbf{b} \odot \mathbf{T}$ (element-wise product with spatial broadcasting) to the final scalar field $\hat{\psi} \in \mathbb{R}^{H \times W}$. While often a simple feed-forward network, we compare MLP and KAN to enhance the model's ability to represent the final solution field.

**Why benchmark on TrunkNet and Regressor?** The selection of architectures is not arbitrary but aims to explore which types of prior knowledge (e.g., spatial multi-scale structures, spectral sparsity, or global relationship dependencies) are most effective for describing the tokamak equilibrium state. The BranchNet handles only low-dimensional parameter encoding, while the TrunkNet and Regressor are responsible for learning the complex spatial structure of solutions.

**Architecture-specific processing of 2D spatial data:**

Below we expand the data flow and inductive biases of each TrunkNet family (KAN/FNO/CNN/Transformer) used in our benchmark. Let $\mathbf{x}_{ij} = (R_i, Z_j)$ denote a grid location and $\mathbf{b} = \mathcal{B}(\xi) \in \mathbb{R}^p$ the BranchNet embedding. The TrunkNet produces a spatial feature map $\mathbf{T} \in \mathbb{R}^{H \times W \times p}$ with per-location feature $\mathbf{t}_{ij} = \mathcal{T}(\mathbf{x}_{ij}) \in \mathbb{R}^p$, and we fuse via element-wise modulation

$$\mathbf{f}_{ij} = \mathbf{b} \odot \mathbf{t}_{ij}, \qquad \hat{\psi}_{ij} = \mathcal{R}(\mathbf{f}_{ij}). \tag{20}$$

This branch–trunk factorization encourages the network to represent *parameter-dependent coefficients* ($\mathbf{b}$) and *spatial basis functions* ($\mathbf{t}_{ij}$) separately, which is particularly effective for operator learning (Lu et al., 2021).

**MLP Trunk (M): point-wise coordinate lifting.** The MLP trunk treats each location independently:

$$\mathbf{t}_{ij} = \phi_\theta\big([\, R_i, Z_j \,]\big), \tag{21}$$

so correlations across space must be learned implicitly through shared weights. This is flexible but can be sample-inefficient for capturing multi-scale global structures.

**KAN Trunk/Regressor (K): learnable 1D spline nonlinearity.** KANs replace fixed activations by learnable 1D functions on edges (Liu et al., 2025). A typical KAN layer can be written as

$$\mathbf{z}_k^{(\ell+1)} = \sum_m \varphi_{km}^{(\ell)}\left(\mathbf{z}_m^{(\ell)}\right), \qquad \varphi_{km}^{(\ell)}(u) = \sum_r c_{kmr}^{(\ell)} B_r(u), \tag{22}$$

where $B_r(\cdot)$ are B-spline basis functions and $c_{kmr}$ are learnable coefficients. In our setting, KAN is used either as the trunk (mapping coordinates to $\mathbf{t}_{ij}$) or as the regressor (mapping $\mathbf{f}_{ij}$ to $\hat{\psi}_{ij}$). Empirically, the adaptive nonlinearity helps fit sharp boundary-layer-like variations of $\psi$ near the last closed flux surface.

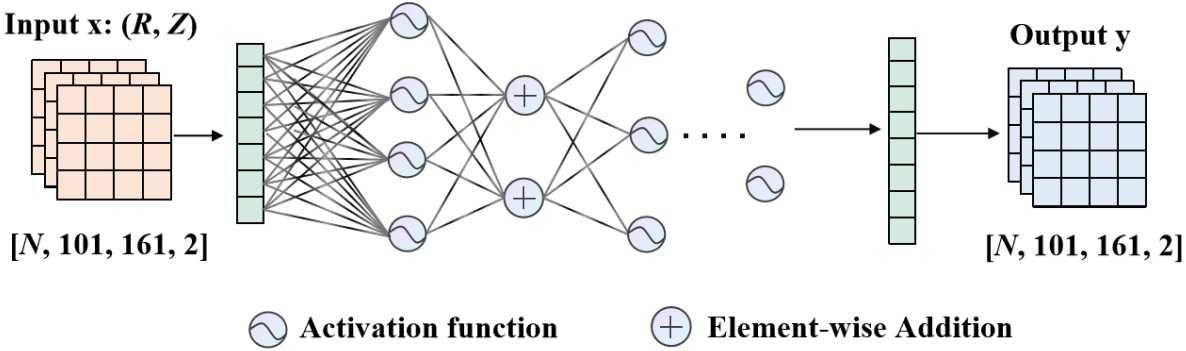

**Activation function** **Element-wise Addition**

*Figure 8.* Schematic of the KAN module (Liu et al., 2025). Each edge applies a learnable 1D spline transform, enabling adaptive nonlinear representation compared to fixed-activation MLPs.

**CNN Trunk (C): local translation-equivariant processing.** CNNs view the coordinate grids as a 2-channel image $\mathbf{X} \in \mathbb{R}^{H \times W \times 2}$ and apply convolutional blocks to produce $\mathbf{T}$. For a kernel $\mathbf{W}$, a convolution layer is

$$[\mathbf{W} * \mathbf{X}]_{ijc} = \sum_{u,v,c'} \mathbf{W}_{uvc'c} \mathbf{X}_{i+u,j+v,c'}. \tag{23}$$

The inductive bias is a *local receptive field* and translation equivariance, which is efficient for local patterns but may struggle to capture global coupling unless depth or dilation is increased.

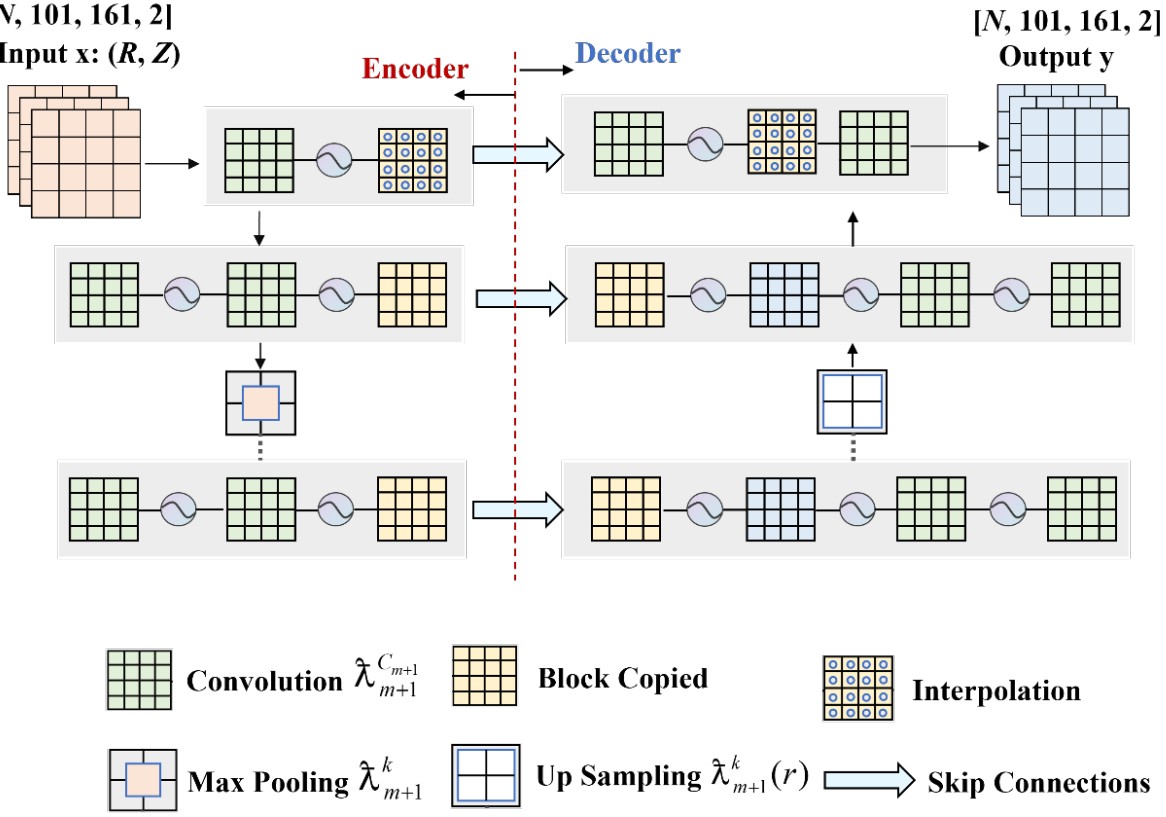

*Figure 9.* Schematic of the CNN trunk. Convolutions build hierarchical local features on the $(R, Z)$ grid and output a spatial feature map for fusion with the BranchNet embedding.

**FNO Trunk (F): global spectral mixing with truncated modes.** FNOs apply learnable spectral convolution (Li et al., 2021). Let $\mathcal{F}$ denote the 2D FFT and $\mathbf{v}^{(\ell)} \in \mathbb{R}^{H \times W \times p}$ the hidden feature map. One FNO layer is

$$\mathbf{v}^{(\ell+1)} = \sigma\Big(\mathbf{W}^{(\ell)}\mathbf{v}^{(\ell)} + \mathcal{F}^{-1}\Big(\mathbf{R}^{(\ell)} \cdot \mathcal{F}(\mathbf{v}^{(\ell)})\Big)\Big), \tag{24}$$

where $\mathbf{R}^{(\ell)}$ contains complex weights on a truncated set of Fourier modes and $\mathbf{W}^{(\ell)}$ is a point-wise linear map. This yields an *effective global receptive field* and tends to model smooth multi-scale fields well.

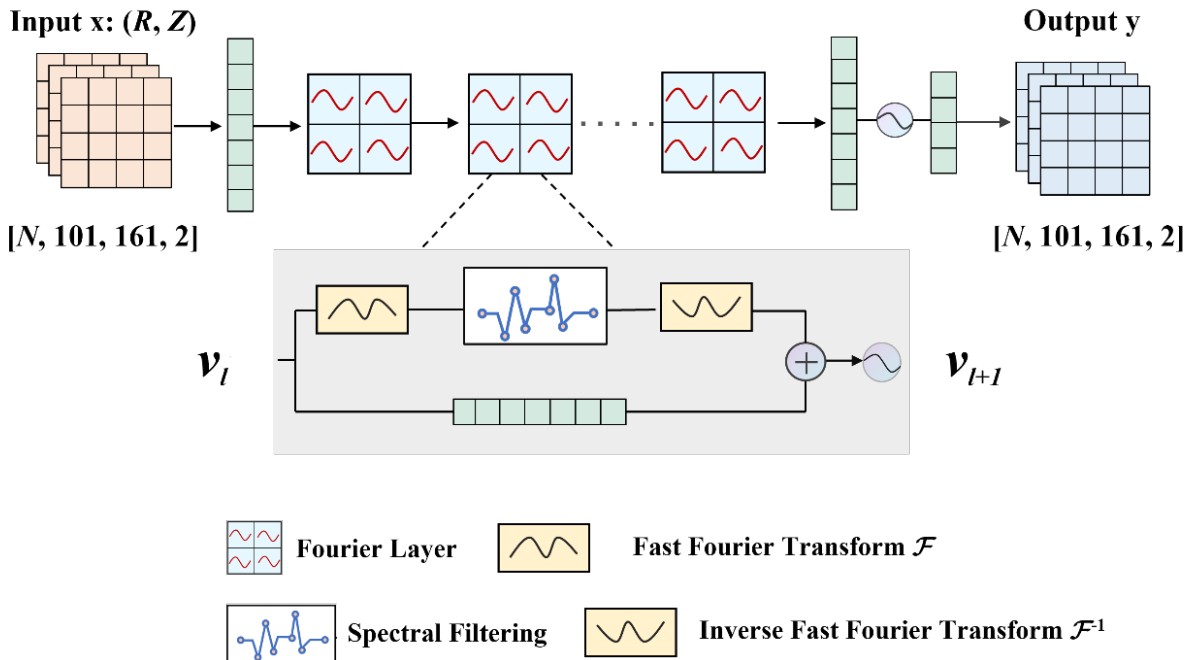

*Figure 10.* Schematic of the FNO trunk (Li et al., 2021). FFT-based spectral convolution mixes global information using learnable weights on a truncated set of modes.

**Transformer Trunk (T): Galerkin global mixing.** The Transformer trunk flattens the grid into $N = H \times W$ tokens $\{\mathbf{h}_n\}_{n=1}^{N}$ with positional embeddings. In the Galerkin/linear attention implementation used by TKNO, each head computes a global feature-coupling matrix rather than an explicit softmax token–token kernel:

$$Y^{(h)} = Q^{(h)}C^{(h)}, \qquad C^{(h)} = \frac{(K^{(h)})^{\top}V^{(h)}}{N}, \tag{25}$$

where $Q^{(h)}, K^{(h)}, V^{(h)}$ are the projected query, key and value features for head $h$. This form aggregates information over all spatial tokens through $C^{(h)}$, providing global mixing without requiring a learned attention matrix to equal a physical Green's kernel. Feed-forward layers and residual connections follow the attention block (Vaswani et al., 2017; Cao, 2021). The resulting global receptive field can capture long-range magnetic-topology-related interactions while keeping the architecture compatible with the FD-based physics loss.

We adopt the naming convention {trunk}–{regressor} with abbreviations M (MLP), K (KAN), F (FNO), C (CNN), T (Transformer) as defined in Section 3.2. For example, T–K denotes a Transformer trunk with a KAN regressor, which we call TKNO.

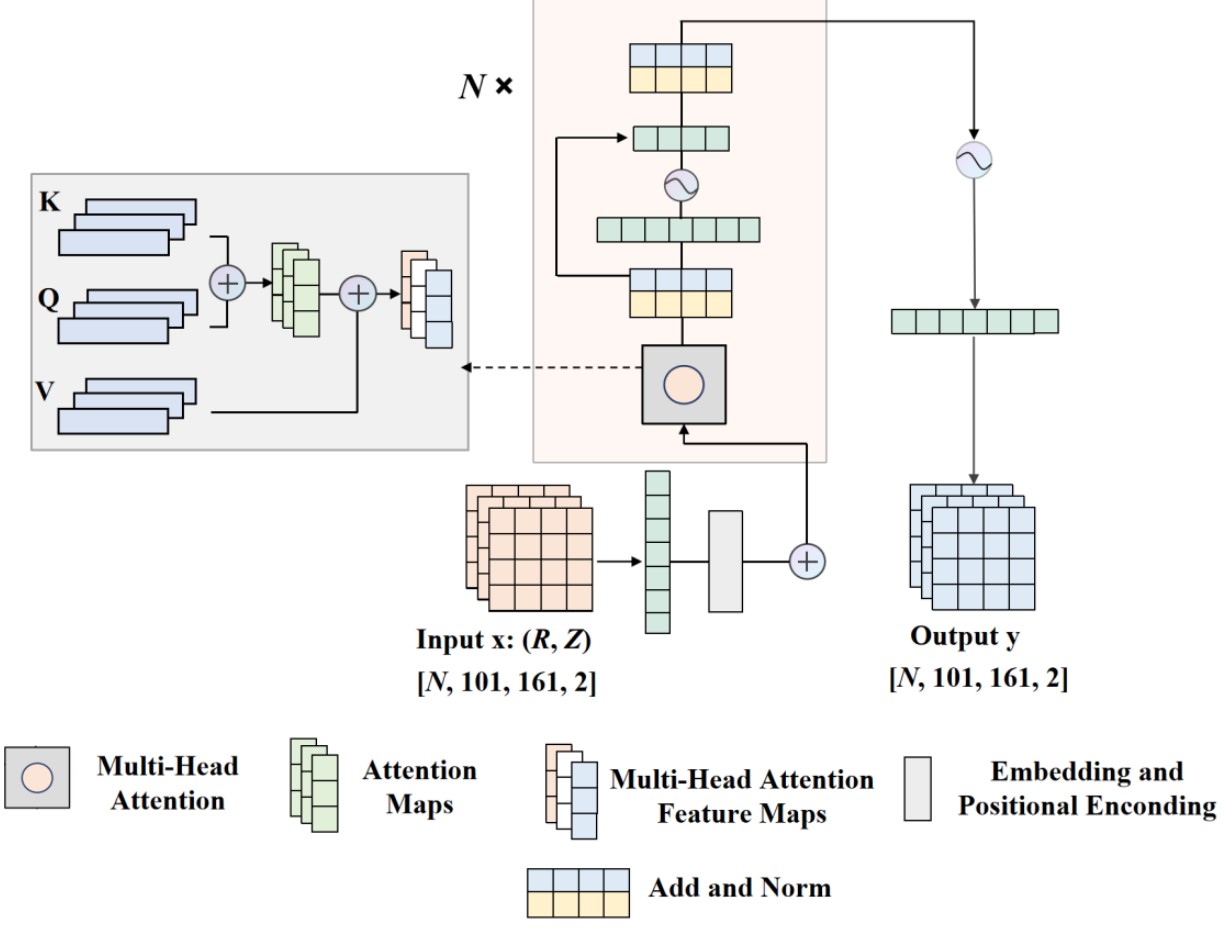

*Figure 11.* Schematic of the Transformer trunk (Vaswani et al., 2017; Cao, 2021). Flattened grid tokens are globally mixed by Galerkin attention and then reshaped back to a spatial feature map.

## B.2. Loss Function

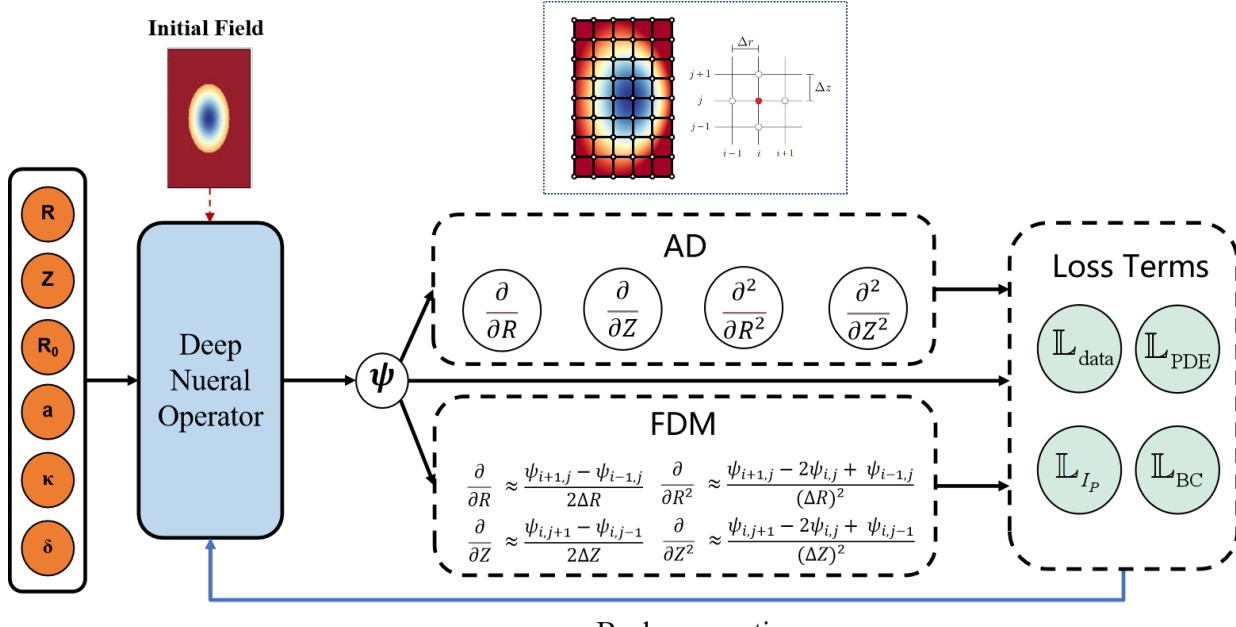

Backpropagation

*Figure 12.* Computation of the composite PINO loss. Data loss uses either full-field supervision or sparse interior anchors; PDE loss uses finite-difference discretization on the output field; boundary and global-constraint losses enforce $\psi|_\Gamma = 0$ and target plasma current $I_p$.

The composite loss for training is:

$$\mathcal{L} = \omega_{\text{data}}\mathcal{L}_{\text{data}} + \omega_{\text{PDE}}\mathcal{L}_{\text{PDE}} + \omega_{\text{BC}}\mathcal{L}_{\text{BC}} + \omega_{I_p}\mathcal{L}_{I_p}. \tag{26}$$

**Data loss** (mean squared error between prediction and ground truth):

$$\mathcal{L}_{\text{data}} = \frac{1}{|\Omega|} \sum_{(i,j)\in\Omega} \left| \hat{\psi}_{ij} - \psi_{ij}^{\text{GT}} \right|^2. \tag{27}$$

For Data-only training, $\mathcal{L}_{\text{data}}$ is computed on the full field. For sparse-anchor variants of Physics-anchored training, $\mathcal{L}_{\text{data}}$ is computed only at $N_s$ randomly sampled points per sample.

**PDE residual** (finite difference; matching the ground-truth solver discretization). The (axisymmetric) Grad–Shafranov operator can be written as

$$\Delta^*\psi \triangleq \frac{\partial^2\psi}{\partial R^2} - \frac{1}{R}\frac{\partial\psi}{\partial R} + \frac{\partial^2\psi}{\partial Z^2}, \tag{28}$$

and the PDE residual at a grid point is $\mathcal{R}_{ij}(\hat{\psi}) = \Delta^*\hat{\psi}(\mathbf{x}_{ij}) + \mu_0 R_i J_\phi(\hat{\psi}_{ij}; \xi)$, where $J_\phi(\cdot)$ follows the prescribed nonlinear profiles in Section 2. Using central differences with grid spacings $\delta R, \delta Z$, we compute

$$\left[\Delta^*\hat{\psi}\right]_{ij} \approx \frac{\hat{\psi}_{i+1,j} - 2\hat{\psi}_{i,j} + \hat{\psi}_{i-1,j}}{(\delta R)^2} - \frac{1}{R_i} \cdot \frac{\hat{\psi}_{i+1,j} - \hat{\psi}_{i-1,j}}{2\delta R} + \frac{\hat{\psi}_{i,j+1} - 2\hat{\psi}_{i,j} + \hat{\psi}_{i,j-1}}{(\delta Z)^2}. \tag{29}$$

We then form the mean-squared residual over interior grid points $\Omega_{\text{int}}$ (inside the plasma mask):

$$\mathcal{L}_{\text{PDE}} = \frac{1}{|\Omega_{\text{int}}|} \sum_{(i,j)\in\Omega_{\text{int}}} \left| \Delta^*_{ij}\hat{\psi} + \mu_0 R_i J_\phi(\hat{\psi}_{ij}) \right|^2. \tag{30}$$

**Domain/masks and sparse anchors.** Let $\Omega$ denote the full $H \times W$ grid, $\Omega_{\text{int}} \subset \Omega$ the interior points inside the plasma boundary, and $\Gamma \subset \Omega$ the discrete boundary (last closed flux surface) where the Dirichlet condition $\psi|_\Gamma = 0$ is imposed.

*Table 8.* Comparison of finite difference (FD) vs. automatic differentiation (AD) for PDE loss computation on compatible architectures ($L_2(\psi)$ in %; MAE(PDE) in original units).

| Method | Derivative | $L_2(\psi)$ | MAE (PDE) |
|--------|-----------|-------------|-----------|
| MLP | FD | **0.11** | **8.35** |
| MLP | AD | 0.30 | 15.37 |
| KAN | FD | **0.06** | **4.29** |
| KAN | AD | 0.17 | 6.46 |

In sparse-anchor variants, we uniformly sample $N_s$ points from the grid per training sample and compute $\mathcal{L}_{\text{data}}$ only on this sparse set, while the PDE/BC/$I_p$ terms are evaluated on the full masks as in Figure 12. In the final calibrated Physics-anchored setting, the same composite objective is used with data-anchor weights chosen to balance numerical scales across data and physics losses.

**Finite Difference vs. Automatic Differentiation:** The PDE residual can be computed using either finite difference (FD) or automatic differentiation (AD). FD approximates derivatives on a discrete grid using neighboring values, while AD computes exact gradients through the computational graph via backpropagation.

- **Finite Difference (FD)**: Compatible with *all* network architectures; matches the discretization of the ground-truth solver; introduces truncation error but provides consistent gradients.

- **Automatic Differentiation (AD)**: Provides exact (machine-precision) derivatives when the model represents $\hat{\psi}(R, Z)$ as a differentiable function of *continuous* coordinates. In our benchmark, this applies naturally to coordinate-based MLP/KAN trunks. For grid-based trunks (CNN/FNO/Transformer), the output is a function of discrete grid tensors and FFT/attention operations, so computing $\partial\hat{\psi}/\partial R$ and $\partial\hat{\psi}/\partial Z$ via coordinate AD is not directly applicable; FD provides a unified alternative.

Table 8 compares the two methods on MLP-based and KAN-based models that support AD.

The results show that FD and AD yield comparable accuracy, but FD is significantly faster due to the overhead of backpropagation through the entire network for AD. Importantly, FD is the only viable option for Transformer- and CNN-based models, enabling physics-informed training across all architectures in our benchmark.

**Boundary condition.** Using the discrete boundary mask $\Gamma$, we enforce the homogeneous Dirichlet condition by

$$\mathcal{L}_{\text{BC}} = \frac{1}{|\Gamma|} \sum_{(i,j)\in\Gamma} \left|\hat{\psi}_{ij}\right|^2. \tag{31}$$

**Plasma current constraint.** The total plasma current is computed by discrete quadrature on $\Omega_{\text{int}}$:

$$I_{p,\text{pred}} \approx \sum_{(i,j)\in\Omega_{\text{int}}} J_\phi(\hat{\psi}_{ij}; \xi)\, \delta R\, \delta Z, \qquad \mathcal{L}_{I_p} = \left|I_{p,\text{pred}} - I_{p,\text{target}}\right|^2. \tag{32}$$

### B.3. Training Configuration

We adopt a unified optimizer and scheduler across experiments, while the effective loss weights are calibrated per paradigm to balance the numerical scales of data and physics terms. Table 9 summarizes the representative TKNO settings used for the main experiments.

## C. Evaluation Metrics

We employ a comprehensive set of evaluation metrics to assess model performance from multiple perspectives.

### C.1. Magnetic Flux Field Prediction Accuracy

This is the most direct assessment of a model's performance, measuring the consistency between the predicted two-dimensional magnetic flux field $\psi_{\text{pred}}$ and the true magnetic flux field $\psi_{\text{true}}$ calculated by a high-precision numerical solver.

*Table 9.* Detailed hyperparameter settings for the TKNO training.

| Category | Parameter | Value |
|---|---|---|
| Optimization | Optimizer | Adam ($\beta_1 = 0.8$, $\beta_2 = 0.9$) |
| | Learning Rate (Initial) | 0.001 |
| | LR Scheduler | MultiStepLR ($\gamma = 0.1$) |
| | | milestones selected per protocol |
| | Random Seed | 2025 |
| Loss Weights | Strategy | fixed |
| | Data-only | data term only |
| | Physics-only | PDE/BC/$I_p$ terms only |
| | Physics-anchored | data + PDE/BC/$I_p$, scale-calibrated |
| Architecture | Input dimension | 10D |
| | Transformer trunk | encoder layers: 2; width: 32 |
| | KAN regressor | layers: 1; width: 64 |

We use two standard mean relative error metrics:

$$\mathcal{L}_{1,\text{rel}}(\psi) = \frac{\|\psi_{\text{pred}} - \psi_{\text{true}}\|_1}{\|\psi_{\text{true}}\|_1}, \tag{33}$$

$$\mathcal{L}_{2,\text{rel}}(\psi) = \frac{\|\psi_{\text{pred}} - \psi_{\text{true}}\|_2}{\|\psi_{\text{true}}\|_2}. \tag{34}$$

The norms are computed over all grid points of each sample and then averaged across the evaluation set.

## C.2. PDE Residual Error

We compute physics consistency on the *predicted* flux field using the same finite-difference (FD) discretization as the numerical solver. Let the discrete residual on the interior mask $\Omega_{\text{int}}$ be

$$r_{ij} = \left[\Delta^*\hat{\psi}\right]_{ij} + \mu_0 R_i J_\phi(\hat{\psi}_{ij}; \xi), \qquad (i,j) \in \Omega_{\text{int}}. \tag{35}$$

We report residual statistics in original units (not %):

$$\text{MAE(PDE)} \triangleq \frac{1}{|\Omega_{\text{int}}|} \sum_{(i,j)\in\Omega_{\text{int}}} |r_{ij}|, \qquad \text{MSE(PDE)} \triangleq \frac{1}{|\Omega_{\text{int}}|} \sum_{(i,j)\in\Omega_{\text{int}}} r_{ij}^2. \tag{36}$$

Note that MSE(PDE) is the *mean squared* residual (not RMSE). These metrics are intentionally distinct from the $\psi$ errors ($L_1(\psi)$, $L_2(\psi)$), which we report as *relative* errors (in %) for accuracy evaluation.

## C.3. Derived Physical Quantities

Beyond direct flux field comparison, we evaluate derived quantities critical for fusion operations:

**Safety Factor ($q$) and $q_{95}$**

The safety factor profile $q(\psi)$ describes the helicity of the magnetic field lines, defined as the limit of the number of toroidal turns a field line makes per poloidal turn. It is a dimensionless measure of the magnetic field line pitch and is topologically invariant:

$$q(\psi) = \frac{f(\psi)}{2\pi} \oint_\psi \frac{dl}{RB_p} \tag{37}$$

where $f(\psi) = RB_\phi$ is the poloidal current function and the line integral is taken along a contour of constant poloidal flux. Of particular importance for stability analysis is the parameter $q_{95}$, defined as the safety factor evaluated at the magnetic flux surface enclosing 95% of the normalized poloidal flux:

$$q_{95} \equiv q(\psi)\big|_{\psi=\psi_{95}} \tag{38}$$

In diverted tokamak configurations, the safety factor diverges to infinity at the separatrix ($q_{\text{sep}} \to \infty$). Therefore, $q_{95}$ serves as a robust representative metric for the edge magnetic topology and is strictly correlated with the stability limits of external kink modes and the onset of disruptions.

**Poloidal Beta ($\beta_p$)**

The poloidal beta measures the efficiency of the poloidal magnetic field in confining the plasma pressure. It is defined as the ratio of the volume-averaged plasma pressure to the magnetic pressure associated with the poloidal magnetic field at the boundary:

$$\beta_p = \frac{2\mu_0 \langle p \rangle_V}{\langle B_p \rangle_a^2} \tag{39}$$

where $\langle p \rangle_V$ is the volume-averaged plasma pressure, and $\langle B_p \rangle_a$ is the poloidal magnetic field averaged over the plasma boundary.

**Normalized Internal Inductance ($l_i$)**

The normalized internal inductance quantifies the magnetic energy stored within the plasma volume relative to the poloidal field energy at the surface. It effectively describes the "peakedness" of the toroidal current density profile:

$$l_i = \frac{\langle B_p^2 \rangle_V}{\langle B_p \rangle_a^2} \tag{40}$$

A higher $l_i$ indicates a more peaked current profile (concentrated near the magnetic axis), while a lower $l_i$ suggests a broader current distribution.

### C.4. EXL-50U Discharge-Data Comparison Metrics

For evaluation on EXL-50U discharge inputs, we compare against the device's standard equilibrium (SE) solver as the operational reference. The numerical values are reported in the main text; here we define the accompanying metrics:

- **RMSE inside LCFS**: Root mean squared error of flux values within the last closed flux surface:

$$\text{RMSE} = \sqrt{\frac{1}{|\Omega_{\text{LCFS}}|} \sum_{(i,j) \in \Omega_{\text{LCFS}}} (\hat{\psi}_{ij} - \psi_{\text{SE},ij})^2} \tag{41}$$

- **Magnetic axis localization error**: Euclidean distance (in mm) between predicted and reference magnetic axis positions.

- **Hausdorff distance**: Maximum deviation (in mm) between predicted and reference last closed flux surfaces, measuring shape fidelity.

- **Shape parameter deviation**: Absolute difference in derived shape parameters, e.g., $|\Delta \kappa|$ for elongation.

# D. Additional Experimental Results

## D.1. Training Dynamics for Data-Only Learning

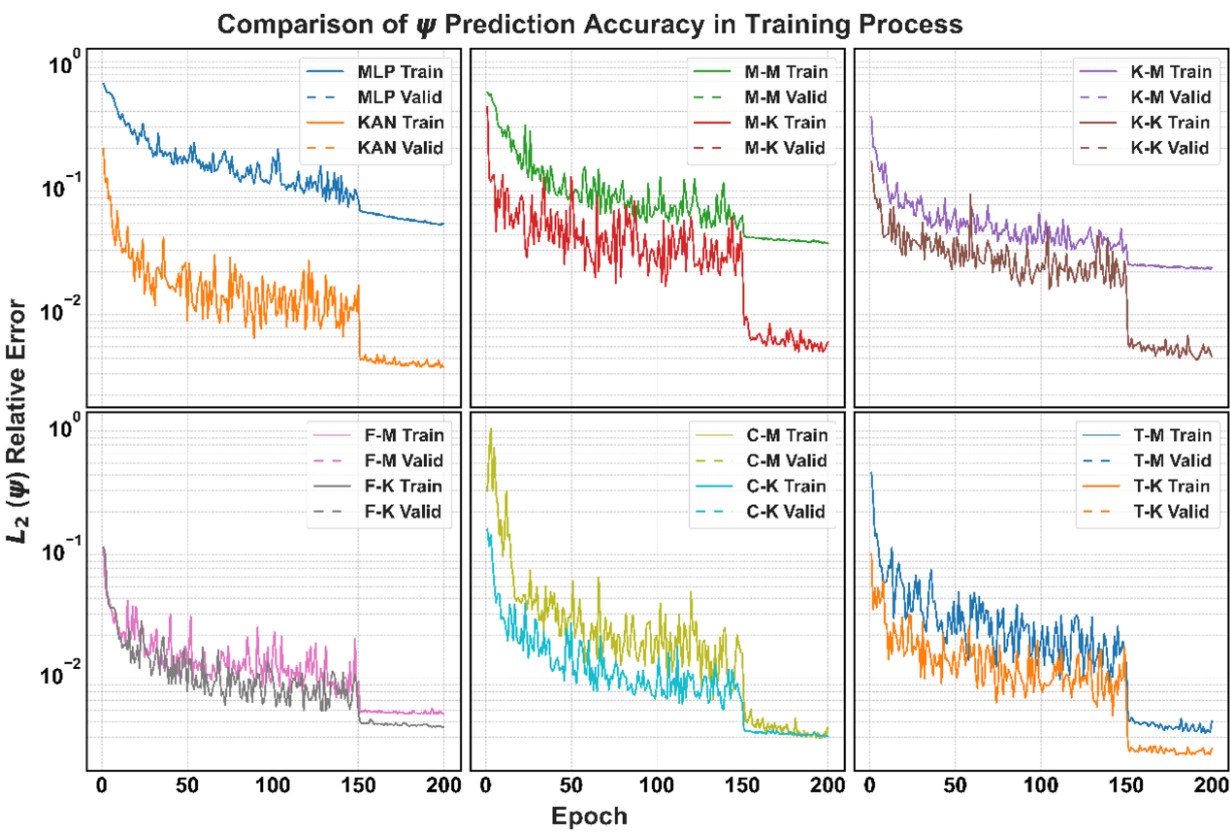

*Figure 13.* Comparison of the $L_2$ relative error for the magnetic flux field $\psi$ on the train and valid datasets, testing standard networks and various NO {trunk net}-{regressor} combinations.

*Table 10.* Full Data-only ablation on GSE: trunk–regressor instantiations and recent baselines on the IID test set. Relative errors are reported as mean $L_2$ in %, while PDE consistency uses mean $L_1$ residual in original units. Flow Matching reports the best U-Net variant.

| Model | Trunk | Regressor | Params (M) | $L_2(\psi)$ | $L_1(\text{PDE})$ | $L_2(I_p)$ | $L_2(q)$ | $L_2(q_{95})$ | $L_2(\beta_p)$ | $L_2(l_i)$ |
|---|---|---|---|---|---|---|---|---|---|---|
| *Pointwise baselines* | | | | | | | | | | |
| MLP | MLP | – | 0.13 | 4.93 | 1.508 | 1.40 | 18.88 | 48.47 | 4742.41 | $2.40\times10^5$ |
| KAN | KAN | – | 0.34 | 0.34 | 2.357 | 0.08 | 1.23 | 2.13 | 0.24 | 0.65 |
| *Trunk–regressor instantiations* | | | | | | | | | | |
| M–M | MLP | MLP | 0.20 | 3.35 | 1.777 | 0.48 | $3.62\times10^7$ | $9.98\times10^7$ | 2.45 | 3.49 |
| K–M | KAN | MLP | 0.47 | 2.11 | 2.042 | 0.45 | $2.63\times10^5$ | $7.31\times10^5$ | 1.10 | 2.52 |
| F–M | FNO | MLP | 0.30 | 0.86 | 3.254 | 0.27 | 4.15 | 9.25 | 0.65 | 1.61 |
| C–M | CNN | MLP | 0.44 | 0.36 | 7.717 | 0.15 | 3.27 | 3.59 | 0.20 | 0.30 |
| T–M | Trans | MLP | 0.28 | 0.40 | 2.712 | **0.06** | 1.06 | 2.37 | 0.36 | 0.72 |
| M–K | MLP | KAN | 0.24 | 0.54 | 2.461 | 0.17 | 1.54 | 1.68 | 0.28 | 0.63 |
| K–K | KAN | KAN | 0.49 | 0.41 | 2.378 | 0.14 | 1.58 | 1.80 | 0.28 | 0.70 |
| F–K | FNO | KAN | 0.30 | 0.36 | 3.158 | 0.07 | 1.22 | 1.37 | 0.31 | 0.74 |
| C–K | CNN | KAN | 0.47 | 0.30 | 5.234 | **0.06** | 1.65 | 2.26 | 0.18 | 0.40 |
| T–K | Trans | KAN | 0.29 | **0.25** | **1.249** | 0.09 | **0.32** | **0.28** | **0.15** | **0.17** |
| *Recent cross-family baselines* | | | | | | | | | | |
| Flow Matching | CNN | – | 0.91 | 0.74 | 8.694 | 0.23 | 3.67 | 7.28 | 0.99 | 1.12 |
| ONO | ONO | – | 0.29 | 0.31 | 1.605 | 0.11 | 0.74 | 1.52 | **0.15** | 0.24 |
| Transolver | Transolver | – | 0.31 | 0.33 | 2.456 | **0.06** | 2.14 | 0.54 | 0.23 | 0.39 |

The training curves show that all models train stably, with consistent performance between their training and validation sets, indicating no significant overfitting or underfitting. The standard MLP (top-left, blue) converges slowly to the highest error, while the KAN baseline (top-left, orange) performs significantly better. The NO models using advanced trunk nets—FNO, CNN, and Transformer (bottom row)—achieve superior performance. Regarding the regressor, KAN also outperforms MLP; however, this improvement becomes marginal when combined with advanced trunk nets. The sharp error drop at epoch 150 results from scheduled learning rate decay.

### D.2. Physics-Only Training Dynamics

To enforce physical consistency, a Physics-only PINO was constructed based on the high-performing TKNO architecture. Since Transformers are not readily compatible with automatic differentiation for computing high-order derivatives, a finite difference method was utilized to compute the derivatives for the PDE loss.

Balancing the loss terms is crucial for successful training, as their gradients can point in conflicting directions, especially given the nonlinear nature of the source term in the GSE. We first tune representative loss weights on controlled cases, then apply the calibrated settings to the 10D dataset used throughout the main text.

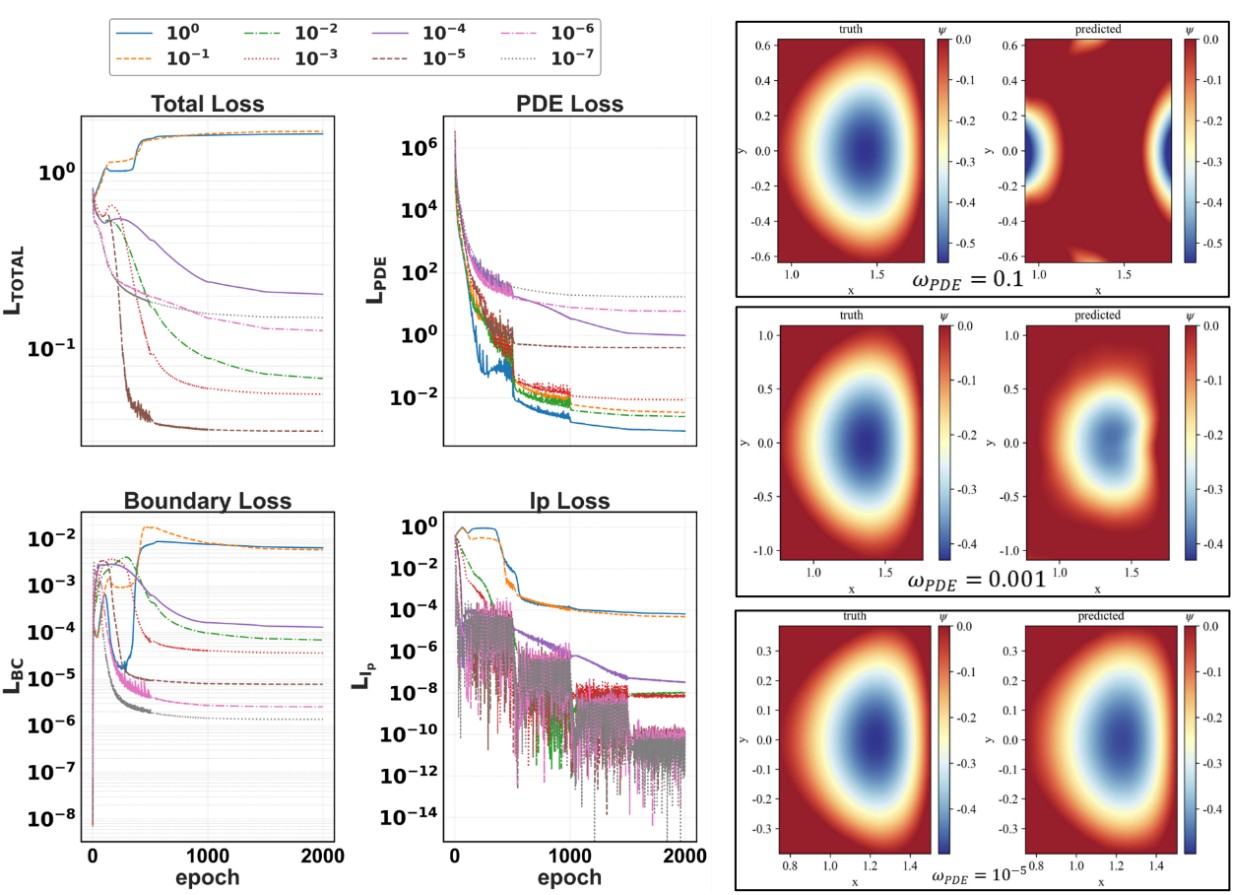

*Figure 14.* PINO training comparison of a single case (the PDE loss weight varies between 1 and 0.0000001, with both boundary and $I_p$ loss weights being 1).

As shown in Figure 14, the weight of the PDE loss term significantly impacts the outcome. A large PDE weight can cause the training to fall into a local minimum that deviates from the true solution, while an appropriately scaled PDE term allows for more balanced optimization. This motivates the scale-calibrated fixed weights used in the main experiments.

## D.3. Detailed OOD Analysis

This subsection expands the main-text OOD evaluation (Table 7) with a finer-grained breakdown of all three training paradigms across the joint and group-isolated OOD pools defined in Section 4.1. Unless otherwise noted, error-based views report $L_2(\psi)$ relative error in %, with the same 1,000-sample IID hold-out baseline as in the main text.

**Large-extrapolation cases in the main text.** Table 11 lists the input parameters for the three qualitative joint-OOD cases shown in Figure 4. Each case contains multiple out-of-range coordinates and is used only as a visual reference for severe extrapolation behavior, not as a representative average sample.

*Table 11.* Input parameters for the large-extrapolation joint-OOD cases in Figure 4.

| Case | $R_0$ | $a$ | $\kappa$ | $\delta$ | $I_p$ | $\beta_0$ | $n_p$ | $m_p$ | $n_f$ | $m_f$ |
|------|-------|------|----------|----------|-------|-----------|-------|-------|-------|-------|
| (a) | 1.99 | 0.50 | 0.98 | -0.39 | $7.17 \times 10^5$ | 0.56 | 2 | 2 | 4 | 4 |
| (b) | 1.86 | 0.23 | 0.68 | 0.56 | $4.96 \times 10^5$ | 0.96 | 2 | 1 | 2 | 1 |
| (c) | 0.86 | 0.22 | 2.53 | 0.24 | $1.65 \times 10^5$ | 0.33 | 1 | 1 | 3 | 2 |

**Distribution and shift-multiplicity views.** Figure 15 gives the boxplot and shift-multiplicity views that complement Table 7 in the main text, using the same isolated and joint OOD subsets as the table. The boxplots show why the mean and tail statistics should be read together: Data-only often has a lower median, while Physics-anchored shortens the right tail on shape-driven and joint shifts. The shift-multiplicity panels show the mean error as more parameters within the same group leave the training range simultaneously.

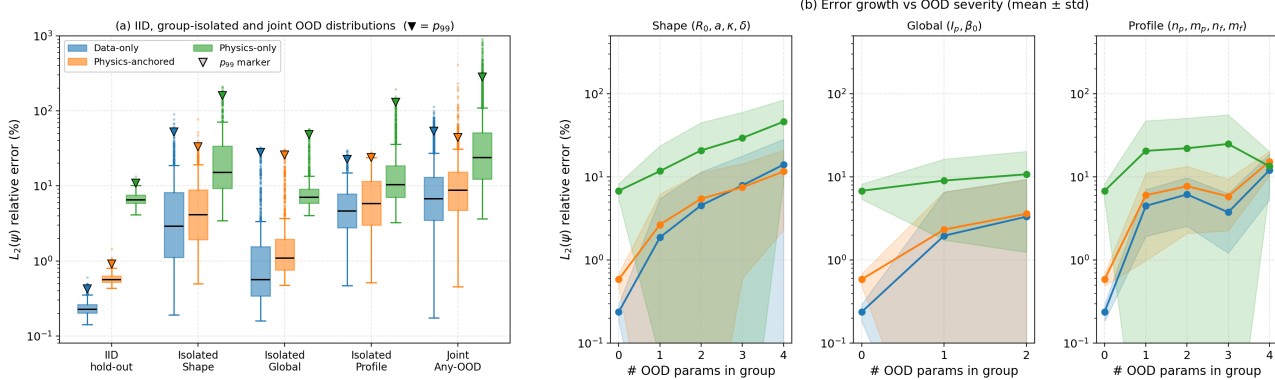

*Figure 15.* Supplementary OOD distribution and shift-multiplicity views using the same OOD subsets as Table 7. **(a)** Boxplots of $L_2(\psi)$ relative error on the IID set, the group-isolated OOD subsets, and the joint Any-OOD pool; downward triangles mark $p_{99}$. **(b)** Mean error versus the number of out-of-range parameters within each group.

**Per-pattern heatmap.** Figure 16 visualizes the mean error of every active OOD pattern within each group (which subset of parameters is simultaneously out of range). Each cell shows the mean $L_2(\psi)$ over the indicated isolated-pool subset, colored on a shared log scale so panels are directly comparable. The pattern view highlights two facts. (i) Within shape, simultaneous multi-parameter shifts such as $a+\kappa$ or $R_0+a+\kappa+\delta$ are among the hardest cases, especially for Physics-only (39–47%), while Data-only and Physics-anchored remain much lower. (ii) For global, shifting only $\beta_0$ is essentially harmless to either Data-only or Physics-anchored (mean $\leq 1\%$) while shifting $I_p$ increases the error, identifying $I_p$ as the dominant global-OOD driver.

**Spectral amplification (full table).** Table 12 accompanies the main-text mechanism analysis (§4.4.1). The error field $e = \hat{\psi} - \psi$ is decomposed in the 2D Fourier domain on the radial frequency grid $k/k_{\max} \in [0, 1]$, with the high-frequency band defined as $k/k_{\max} > 0.35$. The IID hold-out is the same 1000-case split used in Table 7. For computational tractability, spectral diagnostics use a random 1500-case subsample from each joint-OOD slice (Any/Shape/Global/Profile); the table therefore reports spectral error amplification rather than repeating the full-sample $L_2$ growth statistics from Table 7. The high-frequency column is the primary diagnostic: total spectral energy growth is baseline-dependent and should not be read as an absolute accuracy ranking.

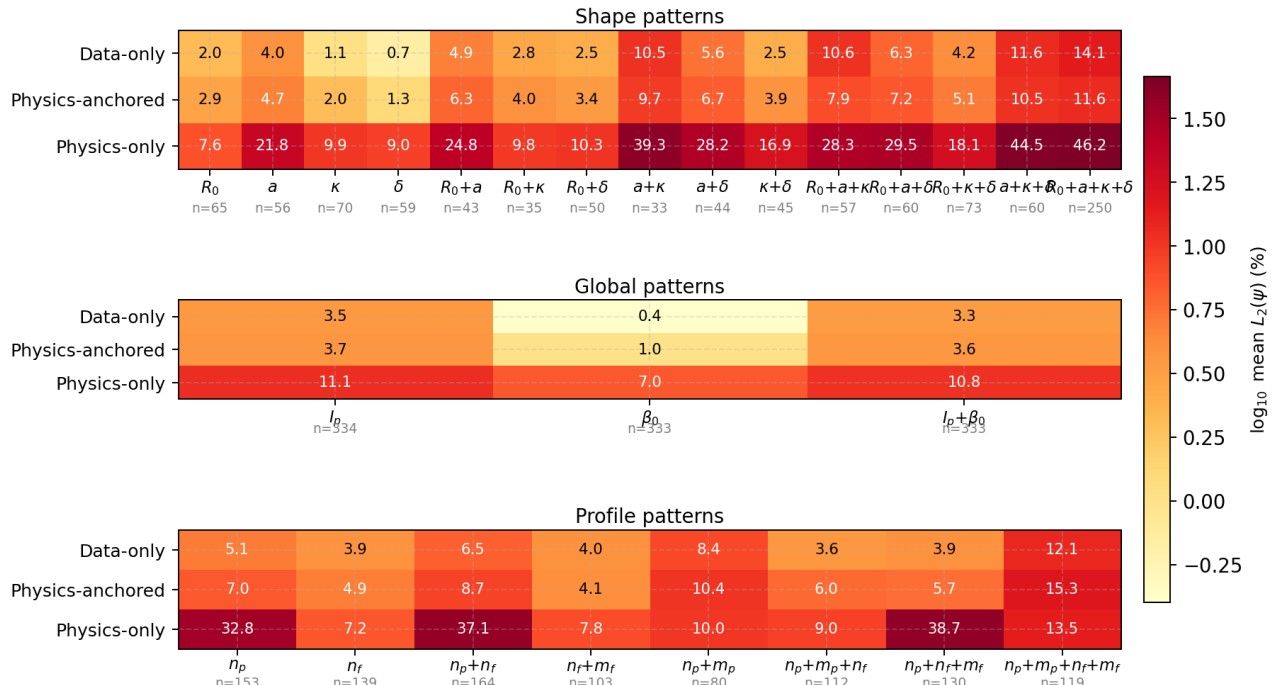

*Figure 16.* Per-pattern mean $L_2(\psi)$ (%) on the isolated OOD pool. Each row is one training paradigm; each column is an active OOD pattern within the indicated group (the value below each label is the sample count). Colors are on a shared $\log_{10}$ scale across panels.

### D.4. Inference Speed

Table 13 shows that TensorRT optimization achieves the best inference speed across all models. For TKNO, TensorRT achieves $\sim$8800$\times$ speedup over the numerical solver (1.16ms vs 10.2s), sufficient for kHz-rate feedback control. The simpler M–K model achieves even faster inference (0.30ms) but with lower accuracy. Note that F–K currently lacks TensorRT support due to complex FFT operations.

## E. Extended Limitations

**Distribution shift scope.** Our OOD evaluation covers shape, global, and profile-parameter shifts, both jointly and in group isolation, together with severity and per-pattern analyses. Other deployment-relevant shifts—including wall/coil conditions, impurity and heating effects, observation noise or missing diagnostics, and discretization/resolution changes—are not systematically studied.

**Fixed-boundary assumption.** Extending to free-boundary equilibrium (coupling to vacuum and coil circuits) is required for closed-loop control applications.

**Static equilibrium assumption.** This work addresses *static* MHD equilibrium, assuming the plasma reaches quasi-steady-state on timescales faster than the control loop. For time-evolving scenarios—where plasma shape and profiles change dynamically during discharge ramp-up/down or transient events—the current framework would require significant extensions:

- **Coupling with transport solvers**: The equilibrium solver could be embedded within a transport simulation loop, where the time evolution of pressure and current profiles is governed by transport equations. At each timestep, the neural operator provides a fast equilibrium update given the evolved profiles.

- **Temporal neural architectures**: Employing neural ODEs, recurrent operators (e.g., LSTM-based), or autoregressive models that can capture the time-dependent dynamics of plasma evolution. This would enable end-to-end learning of the mapping from control inputs and initial conditions to time-evolving equilibria.

*Table 12.* OOD/IID amplification of spectral error energy across shift types (TKNO). The high-frequency column is the primary stability diagnostic.

| Shift | Paradigm | Total spec. E | High-freq. E |
|---|---|---|---|
| Any | Physics-only | $52.5\times$ | $832.6\times$ |
| | Physics-anchored | $855.0\times$ | **7.8×** |
| | Data-only | $6357.8\times$ | $110.6\times$ |
| Shape | Physics-only | $49.7\times$ | $883.1\times$ |
| | Physics-anchored | $856.9\times$ | **7.5×** |
| | Data-only | $5916.8\times$ | $115.7\times$ |
| Global | Physics-only | $48.4\times$ | $872.9\times$ |
| | Physics-anchored | $846.6\times$ | **8.1×** |
| | Data-only | $4998.8\times$ | $83.3\times$ |
| Profile | Physics-only | $59.6\times$ | $1169.0\times$ |
| | Physics-anchored | $965.4\times$ | **7.7×** |
| | Data-only | $5426.1\times$ | $78.4\times$ |

*Table 13.* Comparison of the inference time for different methods.

| Method | CPU (s) | PyTorch (ms) | TorchScript (ms) | TensorRT (ms) |
|---|---|---|---|---|
| FDM-SOR | 10.2 | – | – | – |
| M–K | 0.4 | 0.42 | 0.44 | **0.30** |
| K–K | 4.3 | 13.31 | 2.59 | 0.69 |
| F–K | 3.9 | 6.49 | 4.72 | – |
| C–K | 4.6 | 4.56 | 3.86 | 2.11 |
| T–K | 5.9 | 14.29 | 6.25 | 1.16 |

- **Physics-informed temporal constraints**: Extending the current physics loss to include time-derivative terms from the full MHD equations, enforcing energy conservation and transport physics during dynamic evolution.

Such extensions would enable predictive control applications beyond the current feedforward design paradigm, supporting real-time plasma shape control during transient phases.

**Architecture selection.** TKNO is empirically optimal under our protocol, and our Green-function/Galerkin-attention discussion is intended as design motivation rather than a proof of kernel equivalence. A deeper theoretical characterization of when Galerkin attention and KAN regressors best approximate nonlinear elliptic solution operators remains beyond the scope of this work.

