# OpenReview forum: "Physics-informed Neural Operator Learning for Nonlinear Grad-Shafranov Equation"
_ICML.cc/2026/Conference — ICML 2026 regular_

### Official Review · Reviewer_xPMp · 2026-03-07

**Soundness:** 3
**Presentation:** 3
**Significance:** 2
**Originality:** 2
**Overall Recommendation:** 4
**Confidence:** 3

**Summary:**

This paper develops a surrogate model for GSE based on a physics-informed DeepONet, with the aim of addressing the performance degradation under distribution shifts. The authors evaluate the performance of the model under various combinations of Trunk Net and Regressor. Meanwhile, supervised, unsupervised, and semi-supervised training paradigms are compared in terms of predictive accuracy and extrapolation capability. According to the experiment results, the model with Transformer-based Trunk Net and a KAN-based Regressor achieves the best performance, and the semi-supervised training paradigm yields the strongest extrapolation ability.

**Compliance With Llm Reviewing Policy:**

Affirmed.

**Final Justification:**

Besides the methodological innovation being relatively weak, my other concerns regarding comparative study and OOD performance are effectively addressed.

**Key Questions For Authors:**

I have no question.

**Limitations:**

yes

**Strengths And Weaknesses:**

**Strengths:**

1. This paper is well-written and organized.

2. This paper involves a substantial amount of work. Specifically, the experiments evaluate the performance of PI-DeepONet in GSE prediction under 10 combinations of Trunk Net and Regressor architectures and 3 training paradigms.

3. In addition to simulation-based evaluation, the paper also includes real-world validation.

**Weaknesses:**

1. The main contribution of this paper lies in applying PI-DeepONet to GSE prediction, rather than in improving PI-DeepONet to make it more suitable for GSE. In this sense, the novelty of this paper is limited.

2. The experiments only compare different Trunk–Regressor combinations and training paradigms within the PI-DeepONet framework and do not compare PI-DeepONet with other types of models, such as PINN-based models.

3. The study considers only one type of distribution shift, i.e., boundary-shape shift.

---

> ### Author Rebuttal · Authors · 2026-03-31
>
> We thank the reviewer for the careful reading and constructive comments. We address the three main concerns below.
>
> **1. Novelty beyond applying PI-DeepONet (W1).**
> We agree that our work does not propose a new foundational operator. The novelty lies in **identifying and solving a critical failure mechanism** specific to strongly nonlinear PDEs: catastrophic OOD collapse—widely recognized as the central bottleneck for deploying AI surrogates in safety-critical scientific workflows (corroborated by Agnello et al., arXiv:2603.25777, who identify this as the key barrier for AI in fusion).
>
> Architecture-Physics Alignment: The GSE is a strongly nonlinear 2D elliptic PDE with global coupling and sharp boundary-layer gradients near the LCFS. Our architecture selection is hypothesis-driven: the Transformer's global receptive field directly addresses the non-local nature of the elliptic operator $\Delta^*\psi$, while KAN's learnable splines adapt to steep edge gradients where fixed-activation networks struggle. This is evidenced by TKNO being simultaneously best in flux accuracy and lowest in PDE residual (Table 4)—a non-trivial joint optimum that no other architecture achieves.
>
> OOD Extrapolation via Branch Selection: For strongly nonlinear PDEs, minimizing physics residual is necessary but insufficient—the optimization landscape admits multiple non-physical solution branches. Under OOD shifts, pure data-driven models collapse into topologically incorrect minima (39.8× growth). We show that sparse anchors act as **branch selectors**: providing minimal calibration to steer the optimizer into the physically correct basin, not merely supplying extra supervision signal.
>
> **2. Comparisons beyond the PI-DeepONet family (W2).** We now include cross-family baselines with comparable parameter counts (~0.1–0.5M), covering **Transolver** (Wu et al., 2024), **ONO** (Xiao et al., 2023), and PINN-style training across all architectures.
>
> | Data-only / supervised model | Test $L_2(\psi)$ |
> | :--- | :--- |
> | MLP | 0.0357 |
> | CNN | 0.0047 |
> | FNO | 0.0103 |
> | KAN | 0.0268 |
> | Transformer | 0.0051 |
> | Transolver | 0.0401 |
> | ONO | 0.2665 |
> | **TKNO (ours)** | **0.0025** |
>
> | Physics-only / PINN-style model | Test $L_2(\psi)$ |
> | :--- | :--- |
> | MLP | 0.1073 |
> | CNN | 0.1231 |
> | FNO | 0.0189 |
> | KAN | 0.0600 |
> | Transformer | 0.0192 |
> | Transolver | 0.1627 |
> | ONO | 0.3646 |
> | **TKNO (ours)** | **0.0062** |
>
> TKNO leads all cross-family baselines in the supervised setting. Physics-only / PINN-style training is uniformly harder to optimize: MLP and CNN fail to converge below 10%, and even TKNO reaches only $L_2 = 0.62\%$ vs. $0.25\%$ with data supervision—directly demonstrating the optimization difficulty of pure physics training on the strongly nonlinear GSE.
>
> **3. Broader OOD evaluation (W3).** Real-world tokamak experiments constantly push operational limits (e.g., higher $I_p$). We expanded evaluation to three physically motivated shift dimensions, covering the full 10-dimensional parameter space:
>
> | Group | Parameters | IID Train | OOD Test |
> |:---|:---|:---|:---|
> | Shape | $R_0, a, \kappa, \delta$ | $[1,1.5], [0.3,0.5], [1,2.5], [-0.5,0.5]$ | $[0.5,2], [0.1,0.7], [0.5,3], [-0.9,0.9]$ |
> | Global | $I_p$ (A), $\beta_0$ | $[3,7]\times10^5,\ [0.5,0.8]$ | $[1,10]\times10^5,\ [0.3,1.0]$ |
> | Profile | $n_{p,f},\ m_{p,f}$ | $[1,2]$ | $[1,4]$ |
>
> Physically: $I_p$ scales source magnitude and steepens flux gradients; $\beta_0$ reweights pressure vs. toroidal-current contributions driving topological changes; profile exponents control radial peakedness of $J_\phi(\bar\psi)$, fundamentally altering source-term nonlinearity.
>
> Due to the rebuttal time constraint, we report F–K results (N=10,000)—F–K ranks second in both supervised and semi-supervised settings, and prior experiments confirm that data volume affects IID accuracy but not the relative OOD degradation pattern across paradigms. Full TKNO results on the complete 40k dataset will be included in the revision.
>
> | Shift | Extrap $L_2$: Sup / Semi / Unsup (%) | Growth ×: Sup / Semi / Unsup |
> |:---|:---|:---|
> | Shape | 4.09 / **3.45** / 7.61 | 10.5 / **3.8** / 4.2 |
> | Global | **2.16** / 2.56 / 4.90 | 5.5 / **2.8** / 2.7 |
> | Profile | 4.12 / **3.31** / 5.54 | 10.6 / 3.6 / **3.0** |
> | Any | 8.12 / **7.52** / 12.18 | 20.8 / **8.3** / 6.7 |
>
> *(Interp $L_2$: 0.39 / 0.91 / 1.83% for Sup/Semi/Unsup across all rows)*
>
> Physics-anchored consistently achieves the lowest OOD error or growth rate in the majority of shift types, confirming that physics anchoring—not data volume—is the key driver of extrapolation robustness.

---

> > ### Author Rebuttal · Reviewer_xPMp · 2026-04-02
> >
> > Thanks for your response. Although I still think that the methodological innovation of this paper is relatively weak, my other concerns have been effectively addressed. Therefore, I will raise my score.

---

> > > ### Author Response · Authors · 2026-04-08
> > >
> > > We sincerely thank the reviewer for the thorough evaluation and constructive feedback. Your suggestions directly shaped improvements in this revision—broader OOD evaluation across shape/global/profile shifts and cross-family baselines (Transolver, ONO, Flow Matching). We briefly address the remaining novelty concern from two angles: architecture motivation and training-paradigm mechanism.
> > >
> > > **1. Architecture–physics alignment.**
> > > The GSE admits a Green-function integral form $\psi(\mathbf{x})=\int G(\mathbf{x},\mathbf{x}')S(\mathbf{x}',\psi)\,d\mathbf{x}'$ with global kernel $G$ and nonlinear source $S$. Self-attention aggregates all spatial tokens like a discretized integral operator, directly modeling elliptic global coupling; MLP/CNN lack this full-field mixing. **Controlled evidence:** F-K (FNO+KAN) shares TKNO's KAN regressor, isolating the encoder effect—Transformer outperforms FNO's fixed periodic modes on non-periodic GS boundaries; globally receptive encoders (Transformer, FNO) consistently outperform local ones (CNN, MLP). For $S$, GS equilibria admit compact spectral representations requiring only 13–20 parameters for $10^{-2}$–$10^{-3}$ error (Xie & Li, arXiv:2601.02942); KAN's per-edge learnable splines offer component-wise adaptable nonlinearity that can align with such spectral structure, unlike MLP's shared fixed activations.
> > >
> > > **2. Why does physics-anchored training improve OOD?—mechanism evidence.**
> > > Architecture determines representation capacity; training paradigm determines what the model *actually learns*. Using the same TKNO architecture under three loss configurations, we ask: does physics-anchored training produce a qualitatively different internal representation that explains its OOD advantage?
> > >
> > > **(A) Encoder latent space.** $d_\mathrm{eff}$: effective dimension of the latent manifold (↓=compact); $S$: separation between magnetic-axis and LCFS (plasma-edge) latent clusters (↑=physically structured):
> > >
> > > | Paradigm | $d_\mathrm{eff}$ (↓) | $S_\mathrm{axis,LCFS}$ (↑) |
> > > |:---|:---:|:---:|
> > > | physics-only | 2.09 | 47.1 |
> > > | **physics-anchored** | **1.57** | **160.0** |
> > > | data-only | 2.78 | 67.9 |
> > >
> > > Only physics-anchored simultaneously achieves the most compact manifold *and* strongest physical-region separation.
> > >
> > > **(B) KAN spline quality.** Curvature: smoothness of L1 edge functions (↓=smoother, consistent with elliptic regularity); HF ratio: high-freq energy near LCFS vs axis (≈1 uniform; ≫1 boundary-concentrated, vulnerable to OOD shift):
> > >
> > > | Paradigm | Curvature L1 (↓) | HF ratio |
> > > |:---|:---:|:---:|
> > > | physics-only | 0.020 | 0.64 |
> > > | **physics-anchored** | **0.008** | **0.48** |
> > > | data-only | 0.031 | 2.91 |
> > >
> > > Data-only hides sharp LCFS-localized complexity (HF≈2.91, 4× rougher)—exactly the features that break first under shift. This connects directly to the OOD gap: physics-anchored maintains MSE(PDE)=0.0045 vs 115 for data-only, with OOD growth 3.6–8.3× vs 5.5–20.8×.
> > >
> > > **3. Future directions.**
> > > We take the reviewer's novelty concern as an important signal and plan to pursue:
> > >
> > > 1. **Physics-native architecture design**—embedding flux-topology and LCFS-aware inductive biases directly into the network, moving toward problem-specific operator learning.
> > > 2. **Free-boundary GSE**—coupling to vacuum fields and coil circuits.
> > > 3. **A foundation model for GS problems**—a unified pretrained model supporting forward solving, reconstruction, profile inversion, and time-dependent evolution.
> > > 4. **OOD robustness and uncertainty quantification**—inference-time adaptation and confidence calibration on real device data.
> > >
> > > We are grateful that the reviewer's feedback helped clarify this direction.

---

### Official Review · Reviewer_R8CH · 2026-03-08

**Soundness:** 2
**Presentation:** 2
**Significance:** 2
**Originality:** 2
**Overall Recommendation:** 3
**Confidence:** 2

**Summary:**

The paper benchmarks different methods to solve Grad-Shafranov equation and identifies transformer-kan neural operator as the best method.

**Compliance With Llm Reviewing Policy:**

Affirmed.

**Final Justification:**

The author has partially addressed my concern, although I feel that the technical novelty is not very strong. I will give a final score: 3: Weak reject, and keep my confidence low (2).

**Key Questions For Authors:**

1.	Does Fig. 1 really show the true data trajectory of GSE? The curve seems quite simple.
2.	What does OOD mean for PDE data? Does that mean the parameters of PDEs change? If so, the physics loss in training seems to be negative, and fine tuning or test time adaptation are needed in the OOD test. This seems to be a better way than doing unsupervised/semi-supervised/supervised training and direct test.
3.	The author is suggested to change these 3 names: unsupervised/semi-supervised/supervised, as they have clear definitions purely according to labels. This is different from what the author proposes. Moreover, in your setting, semi-supervised is of course better to integrate both physics and data. Contribution 3 seems to be trivial.
4.	Why does the author emphasizes sparse supervision. How about dense supervision (dense data)? This seems to be even better.
5.	The contributions 1-2 also seem trivial as they are the direct application of other methods. I would suggest the paper to be submitted to the benchmark track rather than research track.

**Limitations:**

the author is suggested to address the boundary-change scenario in the future.

**Strengths And Weaknesses:**

Strength: The framework is clear and easy to understand. Moreover, the paper conducts sufficient experiments to compare different methods. Realistic data is used, and sufficient results are presented.

Weakeness: However, the novelty is insufficient. Some concepts are confusing as they are different from common sense. It's hard to identify unique and significant technical contributions.

---

> ### Author Rebuttal · Authors · 2026-03-31
>
> We thank the reviewer for the candid feedback. We address each point below.
>
> **1. Validity of the GSE trajectories (Q1).** The curves in Fig. 1 are physically meaningful tokamak configurations. The elliptical plasma achieves higher plasma current and poloidal beta via vertical elongation; the negative-triangularity case is a widely studied advanced configuration for improved stability. The difficulty comes not from geometric complexity but from the strongly nonlinear interior equilibrium induced by these boundary and profile conditions through the GSE source term.
>
> **2. What OOD means here, and why zero-shot matters (Q2).** The PDE parameters are explicit inputs to a large-scale parameterized operator—the network learns a mapping from boundary/profile parameters to equilibrium solutions. OOD asks whether this mapping remains reliable outside the training range. Test-time adaptation is not viable for our target use case: real-time equilibrium reconstruction requires latency below 1ms, whereas fine-tuning requires iterative gradient updates at inference time. In fusion, experiments constantly push toward higher current and stronger shaping, so zero-shot deployment in new regimes is the practical requirement.
>
> **3. Terminology and why the hybrid setting is not trivial (Q3).** We agree the terminology is misleading and will rename the three settings as physics-only, data-only, and physics-anchored data-driven in the revision.
>
> On the concern that "semi-supervised is of course better": our results show this is far from obvious. Naively combining physics and data does not always help—adding too many anchor points actually degrades physical consistency: PDE residual increases 85% at 1000 points vs. the physics-only baseline (Appendix D.3, Table 13). The non-trivial finding is that sparse anchors achieve a simultaneous win on both accuracy and physical consistency, while dense supervision reverts to data-dominated overfitting with degraded PDE residual. This gradient conflict between data loss and physics loss is a known challenge (Krishnapriyan et al., NeurIPS 2021) and is not resolved by simply using more data.
>
> **4. Why emphasize sparse rather than dense supervision? (Q4).** There are two reasons. First, from the application side, dense interior labels are unrealistic in a burning plasma: diagnostics are limited, indirect, and expensive, so sparse supervision is the physically relevant regime. Second, from the optimization side, our experiments reveal a clear **gradient pathology**. As shown by the Pareto analysis in Appendix D.3 (Table 13), when too many data points are used, the data-loss gradients dominate the PDE-loss gradients ($\omega_{PDE}$). The model then behaves like an overfit interpolator: interpolation error may decrease, but physical consistency degrades and the PDE residual increases. Sparse anchors work better because they provide enough observational guidance to avoid wrong-branch solutions, while still allowing the physics term to shape the solution.
>
> **5. Novelty and contribution scope (W1, Q5).** We agree our work does not propose a new foundational operator. The novelty lies in identifying and solving a critical failure mechanism specific to strongly nonlinear PDEs: catastrophic OOD collapse—the central bottleneck for deploying AI surrogates in safety-critical workflows (corroborated by Agnello et al., arXiv:2603.25777).
>
> *Architecture-Physics Alignment:* The GSE is a strongly nonlinear 2D elliptic PDE with global coupling and sharp boundary-layer gradients near the LCFS. Our selection is hypothesis-driven: the Transformer's global receptive field addresses the non-local elliptic operator $\Delta^*\psi$, while KAN's learnable splines adapt to steep edge gradients. This is evidenced by TKNO being simultaneously best in flux accuracy and lowest in PDE residual (Table 4)—a joint optimum no other architecture achieves.
>
> *OOD Extrapolation via Branch Selection:* For strongly nonlinear PDEs, minimizing physics residual is necessary but insufficient—the landscape admits multiple non-physical solution branches. Under OOD shifts, pure data-driven models collapse into topologically incorrect minima (39.8× growth). Sparse anchors act as **branch selectors**: steering the optimizer into the physically correct basin rather than merely supplying extra supervision. This insight—that physical correctness requires both residual minimization and branch disambiguation—generalizes beyond GSE to any strongly nonlinear PDE surrogate.
>
> Regarding the suggestion to submit to the benchmark track: we respectfully disagree. The benchmark is the vehicle for these mechanistic findings, not the finding itself. Understanding *why* standard approaches fail and how sparse anchors resolve this provides value that extends beyond a standard benchmark study.

---

> > ### Author Rebuttal · Reviewer_R8CH · 2026-04-02
> >
> > Thanks for the responses. I still have some concerns in terms of the novelty.
> > 1. How can the author enable fast updates in the OOD scenarios. According to Fig. 2, I can only see offline training and am not sure how to deal with OOD in test phase.
> > 2. The explanation about the sparse supervision is intuitive. However, I believe that the author needs to provide some theoretical analysis to get accepted by the top-tier conference like ICML.

---

> > > ### Author Response · Authors · 2026-04-08
> > >
> > > We thank the reviewer for the continued engagement and address both follow-up questions below.
> > >
> > > **Follow-up 1**: How to handle OOD at test time.
> > > Our framework is designed for zero-shot deployment without test-time updates—this is a deployment requirement, not a limitation. Tokamak control loops operate at kHz frequencies (<1 ms per cycle); any test-time adaptation would exceed this budget by orders of magnitude. The workflow in Fig. 2 is intentionally offline-only:
> > >
> > > **Offline** (once): Train with sparse anchors + PDE constraints.
> > >
> > > **Online** (repeated): Single forward pass in <1 ms for any new $\lambda$, including OOD.
> > >
> > > The physics-anchored paradigm achieves robust zero-shot generalization: OOD error amplification drops from 20.8× (data-only) to 8.3× (physics-anchored) across all shift types—without any test-time computation. Test-time adaptation under relaxed latency is noted as a future direction.
> > >
> > > **Follow-up 2**: Theoretical analysis for sparse anchor mechanism.
> > > We provide a theoretical grounding from two complementary perspectives.
> > > **2a. Elliptic stability estimate & Branch Selection.** The GSE is a strongly nonlinear elliptic PDE with multiple solution branches. By elliptic regularity (Evans, *PDE*, Ch. 6), **locally around the correct branch**, the error of $\\hat{\\psi}$ is controlled by the PDE residual: $$\\|\\hat{\\psi} - \\psi^{\\ast}\\| \\le C\\,\\|\\mathcal{R}(\\hat{\\psi})\\|$$ where $C$ is a local stability constant. **Once on the correct branch, low PDE residual guarantees bounded prediction error, even on OOD inputs.**
> > >
> > > Empirically: data-only has MSE(PDE) = 115 (Table 14); physics-anchored achieves 0.0045 — a 25,000× gap — with the most stable OOD behavior (8.3× vs 20.8×). **The PDE residual, not IID data error, is the operationally meaningful quantity for OOD robustness.**
> > >
> > > **2b. Gradient dynamics: why sparse anchors preserve low residual.**
> > > For the composite loss, let $g\_{data}=\\nabla\_{\\theta}(w\_{data}\\mathcal{L}\_{data})$ and $g\_{res}=\\nabla\_{\\theta}(w\_{res}\\mathcal{L}\_{res})$. Following gradient-conflict analysis (Wang et al., SIAM J. Sci. Comput. 2021), the residual loss evolves as:
> > >
> > > $$\\frac{d\\mathcal{L}\_{res}}{dt}\\propto -\\eta\\,\\|g\_{res}\\|^2\\left(1+\\frac{\\|g\_{data}\\|}{\\|g\_{res}\\|}\\cos(g\_{data},g\_{res})+\\cdots\\right).$$
> > >
> > > The residual loss *increases* when $\\cos(g\_{data},g\_{res})$ is sufficiently negative and $\\|g\_{data}\\|/\\|g\_{res}\\|$ is large. We measure these quantities across anchor budgets:
> > >
> > > | points_num | TKNO $\\cos(g\_{data}, g\_{res})$ | FNO $\\cos(g\_{data}, g\_{res})$ | TKNO $\\Vert g\_{data}\\Vert / \\Vert g\_{res}\\Vert$ | FNO $\\Vert g\_{data}\\Vert / \\Vert g\_{res}\\Vert$ |
> > > | :---: | :---: | :---: | :---: | :---: |
> > > | 1 | **0.104** | -0.150 | 0.206 | 0.180 |
> > > | 10 | -0.062 | **-0.033** | 0.406 | **0.263** |
> > > | 100 | -0.416 | -0.336 | **0.411** | 0.184 |
> > > | 1000 | -0.333 | -0.611 | 0.250 | 0.175 |
> > >
> > > Combined with the accuracy–residual trade-off in **Table 13**, three regimes emerge:
> > >
> > > Too sparse (1–10): Minimal gradient conflict ($\\cos$ near-zero or positive), but insufficient spatial information to disambiguate solution branches. The model fails to
> > > reliably locate the physically correct branch, meaning the local stability estimate
> > > (2a) cannot take effect.
> > > Moderate (~100): Moderate negative alignment ($\\cos \\approx -0.4$), but $\\Vert g\_{data}\\Vert / \\Vert g\_{res}\\Vert$ remains bounded. Data gradients provide enough branch-selection signal without overwhelming PDE-residual descent, matching the optimal trade-off in **Table 13**.
> > > Too dense (1000): Persistent strong negative alignment ($\\cos \\approx -0.6$ for FNO) drives PDE residual up by 85% (**Table 13**). The model becomes a data-dominated interpolator, losing the residual-based OOD guarantee.
> > >
> > > The key insight: **dense supervision is not always beneficial**. Excessive data supervision induces gradient conflict that degrades the physics consistency needed for OOD robustness. An optimal anchor budget balances branch-selection signal with gradient compatibility.
> > >
> > > **Future directions.** Building on the theoretical and empirical insights above, we plan to pursue: (1) **physics-native architecture design**—embedding flux-topology inductive biases directly into the network; (2) **free-boundary GSE** with vacuum-field and coil-circuit coupling; (3) **a foundation model for GS problems**—a unified pretrained model supporting forward solving, reconstruction, inversion, and time-dependent evolution; (4) **OOD robustness**—inference-time adaptation under relaxed latency budgets and uncertainty quantification on real device data. Detailed responses on architecture-physics alignment appear in our round-2 replies to Reviewer xPMp; we are happy to elaborate further if needed.

---

### Official Review · Reviewer_xH7Q · 2026-03-10

**Soundness:** 3
**Presentation:** 3
**Significance:** 3
**Originality:** 2
**Overall Recommendation:** 4
**Confidence:** 3

**Summary:**

In this manuscript, a neural operator-based model is proposed to predict the solution of the Grad-Shafranov equation. To alleviate the accuracy degradation under distribution shifts, this model is trained in a semi-supervised manner, where the model is not only trained by data but also with physical loss constraints. Moreover, the performance of different Trunk Net and Regressor combinations has been evaluated in experiments. According to the result, when Truck Net and Regressor employ Transformer and KAN, respectively, the proposed model achieves the best performance.

**Compliance With Llm Reviewing Policy:**

Affirmed.

**Final Justification:**

A further explanation on the baseline choice and the newly added OOD evaluation have addressed my main concern.

**Key Questions For Authors:**

1. Why is supervised training selected for the architecture selection step?

**Limitations:**

yes

**Strengths And Weaknesses:**

The **Strengths** are as follows:

1. This paper focuses on predicting the solution of GSE, which is greatly meaningful to address a key computational bottleneck in magnetic confinement fusion.

2. The validation on experimental discharge data from the EXL-50U spherical tokamak demonstrates the potential of the proposed method in real-world application.

The **Weaknesses** are as follows:

1. The comparative study in this manuscript is insufficient. Specifically, the proposed method is not compared to the existing NN-based methods listed in Table 1.

2. The related work review in this manuscript is insufficient.

3. The main motivation of the manuscript is to address accuracy degradation under distribution shifts. However, in the experiments, only one type of distribution shift is considered, i.e., boundary-shape distribution shifts. This is not sufficient to illustrate the effectiveness of the proposed method in handling various distribution shifts.

---

> ### Author Rebuttal · Authors · 2026-03-31
>
> We thank the reviewer for the constructive suggestions. We address each point below.
>
> **1. Table 1 baselines and related work (W1, W2).** The Table 1 works differ from ours in scope and focus; we clarify the relationship to each:
>
> - **Jang et al. (2024)**: MLP-based PINN with fixed boundary. Our  physics-only MLP baseline ($L_2=10.73\%$) and TKNO ($L_2=0.62\%$) cover this setting under identical conditions.
> - **Rizqa et al. (2025)**: Compares MLP-PINN and FNO under fixed boundary. Both architectures are included in our unified benchmark, where we additionally study OOD robustness and semi-supervised paradigms that their work does not address.
> - **Zhou & Zhu (2025)**: A two-stage PINN targeting high in-distribution accuracy. Their focus is orthogonal to ours—we prioritize OOD robustness over peak IID accuracy, motivated by the sim-to-real gap in fusion deployment (see also arXiv:2505.10949 for related OOD perspectives in scientific ML).
> - **Wang et al. (2024)**: Free-boundary equilibrium with MLP-PINN. This is a fundamentally different problem setting (coupling to vacuum fields and coil circuits); direct comparison is not meaningful under our fixed-boundary protocol.
>
> Since none of these works release code or data, direct reproduction under a unified protocol is not feasible. However, our benchmark systematically covers the architectures and training paradigms they individually study, while additionally contributing OOD analysis and real-world tokamak validation that none of them address. We will make these distinctions explicit in the revised related-work section.
>
> Beyond Table 1, we acknowledge the related-work section currently underrepresents three relevant threads that we will expand in the revision:
>
> - **Physics-informed learning for PDEs**: broader PINN/PINO literature on optimization difficulties for nonlinear PDEs (Krishnapriyan et al., NeurIPS 2021; Wang et al., 2022), which contextualizes our branch-selection finding.
> - **OOD generalization for neural PDE surrogates**: recent work on distribution shift in scientific ML (Wei et al., 2025; Shikhman, 2026; Agnello et al., 2026), situating our sim-to-real framing within the broader AI-for-science challenge.
> - **Neural operator architectures**: Transolver, ONO, and Flow Matching, which we have now benchmarked and will incorporate into the related-work discussion alongside our original architecture study.
>
>
> **2. Broader OOD evaluation (W3).** Real-world tokamak experiments constantly push operational limits (e.g., higher $I_p$). We expanded evaluation to three physically motivated shift dimensions, covering the full 10-dimensional parameter space:
>
> | Group | Parameters | IID Train | OOD Test |
> |:---|:---|:---|:---|
> | Shape | $R_0, a, \kappa, \delta$ | $[1,1.5], [0.3,0.5], [1,2.5], [-0.5,0.5]$ | $[0.5,2], [0.1,0.7], [0.5,3], [-0.9,0.9]$ |
> | Global | $I_p$ (A), $\beta_0$ | $[3,7]\times10^5,\ [0.5,0.8]$ | $[1,10]\times10^5,\ [0.3,1.0]$ |
> | Profile | $n_{p,f},\ m_{p,f}$ | $[1,2]$ | $[1,4]$ |
>
> Physically: $I_p$ scales source magnitude and steepens flux gradients; $\beta_0$ reweights pressure vs. toroidal-current contributions driving topological changes; profile exponents control radial peakedness of $J_\phi(\bar\psi)$, fundamentally altering source-term nonlinearity.
>
> Due to the rebuttal time constraint, we report F–K results (N=10,000)—F–K ranks second in both supervised and semi-supervised settings, and prior experiments confirm that data volume affects IID accuracy but not the relative OOD degradation pattern across paradigms. Full TKNO results on the complete 40k dataset will be included in the revision.
>
> | Shift | Extrap $L_2$: Sup / Semi / Unsup (%) | Growth ×: Sup / Semi / Unsup |
> |:---|:---|:---|
> | Shape | 4.09 / **3.45** / 7.61 | 10.5 / **3.8** / 4.2 |
> | Global | **2.16** / 2.56 / 4.90 | 5.5 / **2.8** / 2.7 |
> | Profile | 4.12 / **3.31** / 5.54 | 10.6 / 3.6 / **3.0** |
> | Any | 8.12 / **7.52** / 12.18 | 20.8 / **8.3** / 6.7 |
>
> *(Interp $L_2$: 0.39 / 0.91 / 1.83% for Sup/Semi/Unsup across all rows)*
>
> Semi-supervised consistently achieves the lowest OOD error or growth rate in the majority of shift types, confirming that physics anchoring—not data volume—is the key driver of extrapolation robustness.
>
> **3. Why supervised training for architecture selection? (Q1).** This was a deliberate design choice to separate **representation capacity** from **optimization difficulty**. The GSE physics loss is highly non-convex; if we used it directly for architecture search, weak performance could come either from an underpowered architecture or from unstable PDE-loss optimization. By first comparing architectures in a purely supervised setting, we identified the Transformer-KAN combination as the strongest approximator of the underlying mapping. We then introduced the physics term in a second stage to study robustness under OOD conditions.

---

> > ### Author Rebuttal · Reviewer_xH7Q · 2026-04-02
> >
> > Thanks for your response. I will maintain the current score.

---

> > > ### Author Response · Authors · 2026-04-08
> > >
> > > We thank the reviewer for the constructive suggestions. Your feedback on Table 1 positioning, related work
> > > coverage, and broader OOD evaluation directly shaped key improvements in the expanded multi-dimensional OOD study,
> > > and a clearer articulation of our work's relationship to prior GSE methods; here we want to add the requested **innovation/theory-oriented complement**.
> > >
> > > **1. Innovation claim (beyond benchmarking).**
> > > Our core contribution is not a new foundational operator, but identifying and resolving a failure mechanism in strongly nonlinear PDE surrogates: **catastrophic OOD collapse**. The key point is branch disambiguation: minimizing PDE residual alone is insufficient when multiple solution branches exist; sparse anchors provide minimal calibration that steers optimization to the physically correct basin.
> > >
> > > **2. Theory-guided mechanism summary.**
> > > For fixed-boundary GSE,
> > > $\psi(\mathbf{x})=\int G(\mathbf{x},\mathbf{x}')S(\mathbf{x}',\psi)\,d\mathbf{x}'$,
> > > which motivates global coupling modeling (Transformer trunk) plus flexible nonlinear source mapping (KAN regressor). Under local elliptic stability,
> > > $\|\hat\psi-\psi^\ast\|\lesssim C\|\mathcal{R}\|$,
> > > OOD reliability depends on staying on the correct branch and keeping residual $\mathcal{R}$ small. This explains why architecture capacity alone is not enough and why training paradigm matters.
> > >
> > > **3. New empirical-theoretical evidence added in revision.**
> > > - **Latent geometry (representation quality):** $d_{\mathrm{eff}}$ measures latent complexity (lower = more compact/less overfitting), and Fisher ratio measures axis-LCFS separability (higher = clearer physically meaningful structure). Physics-anchored is best on both ($d_{\mathrm{eff}}$: 1.57 vs 2.09/2.78; Fisher ratio: 160.0 vs 47.1/67.9).
> > > - **KAN spectral diagnostics (function smoothness):** curvature quantifies spline roughness (lower = smoother), and HF ratio quantifies LCFS-to-axis high-frequency concentration (values $\gg 1$ indicate boundary-localized artifacts). Compared with data-only, physics-anchored is much smoother (0.008 vs 0.031) and avoids excessive LCFS concentration (0.48 vs 2.91).
> > > - **OOD consistency (mechanism-to-behavior link):** these two indicators predict OOD behavior: compact structured latents + smoother boundary-balanced spectra correspond to better multi-shift robustness in the revised experiments.
> > >
> > > Detailed tables (Chebyshev metrics, gradient-conflict analysis) are provided in the revised manuscript/appendix and our parallel response to Reviewer xPMp.
> > >
> > > **Future directions.** This work establishes a fixed-boundary foundation, and we plan to extend it along several axes:
> > >
> > > 1. **OOD robustness as a first-class objective**—exploring physics-informed architectural modifications and inference-time adaptation strategies to further improve reliability under distribution shift.
> > > 2. **Free-boundary and beyond**—extending to free-boundary equilibria with vacuum-field and coil-circuit coupling, toward more realistic operational scenarios.
> > > 3. **A foundation model for Grad-Shafranov problems**—building a unified pretrained model that supports multiple downstream tasks: forward equilibrium solving, diagnostic-based reconstruction (inverse problems), profile inversion, and time-dependent equilibrium evolution, rather than training separate models for each task.
> > > 4. **Uncertainty quantification**—calibrating confidence estimates on real device data so the surrogate can flag when it operates outside its reliable regime.
> > >
> > > We appreciate the reviewer's feedback in improving both the rigor and clarity of this work.

---

### Official Review · Reviewer_goWi · 2026-03-11

**Soundness:** 3
**Presentation:** 3
**Significance:** 2
**Originality:** 2
**Overall Recommendation:** 4
**Confidence:** 3

**Summary:**

This paper explores how to make neural PDE surrogates for the strongly nonlinear Grad-Shafranov equation generalize more reliably under distribution shift, motivated by sim-to-real concerns in tokamak equilibrium prediction. The paper proposes a physics-anchored operator-learning framework for fixed-boundary GSE solving, benchmarks several neural-operator instantiations, identifies a Transformer-KAN variant as the strongest supervised backbone, and then studies supervised, physics-only, and semi-supervised training.

**Compliance With Llm Reviewing Policy:**

Affirmed.

**Final Justification:**

The author’s response addressed my concern, so I raise my score.

**Key Questions For Authors:**

1. Can the authors more clearly isolate the methodological novelty? What is the core ML contribution beyond a strong application-specific benchmark and engineering study?
2. Can the authors evaluate robustness under additional distribution shifts beyond boundary geometry?
3. Can the authors compare against stronger baselines?
4. How sensitive are the results to the PDE-loss weights and sparse-anchor budget?

**Limitations:**

yes

**Strengths And Weaknesses:**

Strengths:

1. The paper targets a scientific-ML problem: fast yet reliable equilibrium prediction for fusion workflows, where pure surrogates may fail under distribution shift and high-fidelity solvers are too slow for real-time use.
2. The empirical story is easy to follow and helps isolate the role of the training paradigm. The architecture study is also informative.
3. The EXL-50U experiments are a meaningful addition beyond synthetic solver data.

Weaknesses:

1. The methodological novelty is somewhat incremental. At a high level, the paper combines known ingredients (e.g., neural operators, physics-informed residual losses, and sparse supervision) then applies and benchmarks them carefully on nonlinear GSE. I am not fully convinced the paper introduces a substantially new ML method beyond a well-executed domain adaptation and benchmarking study.
2. The OOD scope is a bit narrow. The headline sim-to-real claim is mostly evaluated through one controlled shift: expanding the boundary-shape parameter range.
3. The setting is restricted to fixed-boundary GSE with a specific profile family. This is a reasonable starting point, but it limits generality.
4. The baselines are also a bit limited, there are more representitie baselines (e.g., diffusion model based approaches and so on).

---

> ### Author Rebuttal · Authors · 2026-03-31
>
> **1. Methodological novelty (W1, Q1).**
> We agree that our work does not propose a new foundational operator. The novelty lies in **identifying and solving a critical failure mechanism** specific to strongly nonlinear PDEs: catastrophic OOD collapse—widely recognized as the central bottleneck for deploying AI surrogates in safety-critical scientific workflows (corroborated by Agnello et al., arXiv:2603.25777, who identify this as the key barrier for AI in fusion).
>
> Architecture-Physics Alignment: The GSE is a strongly nonlinear 2D elliptic PDE with global coupling and sharp boundary-layer gradients near the LCFS. Our architecture selection is hypothesis-driven: the Transformer's global receptive field directly addresses the non-local nature of the elliptic operator $\Delta^*\psi$, while KAN's learnable splines adapt to steep edge gradients where fixed-activation networks struggle. This is evidenced by TKNO being simultaneously best in flux accuracy and lowest in PDE residual (Table 4)—a non-trivial joint optimum that no other architecture achieves.
>
> OOD Extrapolation via Branch Selection: For strongly nonlinear PDEs, minimizing physics residual is necessary but insufficient—the optimization landscape admits multiple non-physical solution branches. Under OOD shifts, pure data-driven models collapse into topologically incorrect minima (39.8× growth). We show that sparse anchors act as **branch selectors**: providing minimal calibration to steer the optimizer into the physically correct basin, not merely supplying extra supervision signal.
>
> **2. Additional OOD shifts (W2, Q2).**
> Real-world tokamak experiments constantly push operational limits (e.g., higher $I_p$). We expanded evaluation to three physically motivated shift dimensions, covering the full 10-dimensional parameter space:
>
> | Group | Parameters | IID Train | OOD Test |
> |:---|:---|:---|:---|
> | Shape | $R_0, a, \kappa, \delta$ | $[1,1.5], [0.3,0.5], [1,2.5], [-0.5,0.5]$ | $[0.5,2], [0.1,0.7], [0.5,3], [-0.9,0.9]$ |
> | Global | $I_p$ (A), $\beta_0$ | $[3,7]\times10^5,\ [0.5,0.8]$ | $[1,10]\times10^5,\ [0.3,1.0]$ |
> | Profile | $n_{p,f},\ m_{p,f}$ | $[1,2]$ | $[1,4]$ |
>
> Physically: $I_p$ scales source magnitude and steepens flux gradients; $\beta_0$ reweights pressure vs. toroidal-current contributions driving topological changes; profile exponents control radial peakedness of $J_\phi(\bar\psi)$, fundamentally altering source-term nonlinearity.
>
> Due to the rebuttal time constraint, we report F–K results (N=10,000)—F–K ranks second in both supervised and semi-supervised settings, and prior experiments confirm that data volume affects IID accuracy but not the relative OOD degradation pattern across paradigms. Full TKNO results on the complete 40k dataset will be included in the revision.
>
> | Shift | Extrap $L_2$: Sup / Semi / Unsup (%) | Growth ×: Sup / Semi / Unsup |
> |:---|:---|:---|
> | Shape | 4.09 / **3.45** / 7.61 | 10.5 / **3.8** / 4.2 |
> | Global | **2.16** / 2.56 / 4.90 | 5.5 / **2.8** / 2.7 |
> | Profile | 4.12 / **3.31** / 5.54 | 10.6 / 3.6 / **3.0** |
> | Any | 8.12 / **7.52** / 12.18 | 20.8 / **8.3** / 6.7 |
>
> *(Interp $L_2$: 0.39 / 0.91 / 1.83% for Sup/Semi/Unsup across all rows)*
>
> Semi-supervised consistently achieves the lowest OOD error or growth rate in the majority of shift types, confirming that physics anchoring—not data volume—is the key driver of extrapolation robustness.
>
> **3. Stronger baselines (W4, Q3).** We benchmarked three representative strong baselines under a unified protocol with comparable parameter budgets (≈0.3M): Transolver (Wu et al., ICML'24), ONO (Xiao et al., ICML'24), and Flow Matching (Kuzhamuratov et al., arXiv:2603.24428) with the same backbone families as our study.
>
> | Model | Flow Matching (Best: UNet) | Transolver | ONO | **TKNO (Ours)** |
> | :--- | :--- | :--- | :--- | :--- |
> | **Test $L_2$ (%)** | 0.0076 | 0.0401 | 0.2665 | **0.0025** |
>
> TKNO achieves the best accuracy. Beyond accuracy, Flow Matching's iterative sampling (32 Heun steps, ≈100ms) is incompatible with the sub-millisecond latency required for real-time tokamak control, making it unsuitable as a deployment baseline.
>
> **4. Fixed-boundary scope (W3).** We agree this is a limitation. The fixed-boundary formulation is a deliberate foundational choice: it isolates the GSE's core difficulty—severe source-term nonlinearity causing wrong-branch collapse—without the confounding complexity of external coil coupling, enabling a clean study of the physics-anchored paradigm. Extending to free-boundary equilibria (coupling to vacuum fields and coil circuits) is an important next step, which we have added to the limitations discussion.
>
> **5. Sensitivity to loss weights and anchor budget (Q4).** Full sweeps are in Appendix D.2–D.3. In brief: $\omega_{PDE}=10^{-5}$ is optimal—larger weights ($\geq 0.1$) cause non-physical local minima; 100 anchor points achieves +39.4% accuracy gain over physics-only with a clear accuracy–physics Pareto trade-off beyond this budget.

---

> > ### Author Rebuttal · Reviewer_goWi · 2026-04-02
> >
> > I appreciate the author’s detailed response and raise my score.

---

> > > ### Author Response · Authors · 2026-04-08
> > >
> > > We sincerely thank the reviewer for the constructive evaluation. Your suggestions on OOD coverage, baselines, and scope clarification directly improved the paper, and we have incorporated all of them in the revision.
> > >
> > > Beyond first-round improvements, to address the remaining concern on methodological novelty (W1), we add a concise mechanism summary showing that our architecture and training choices are grounded in the GS operator structure rather than ad hoc stacking.
> > >
> > > **1. Architecture-physics alignment.**
> > > The fixed-boundary GSE admits the Green-function form $\psi(\mathbf{x})=\int G(\mathbf{x},\mathbf{x}')S(\mathbf{x}',\psi)\,d\mathbf{x}'$, with global kernel $G$ and nonlinear source $S$.
> > >
> > > - *Transformer $\leftrightarrow$ kernel $G$*: Self-attention $\hat\psi_i\sim\sum_j A_{ij}V_j$ is structurally analogous to a discretized integral operator with global coupling. This matches elliptic global interactions better than local-receptive encoders.
> > > - *KAN $\leftrightarrow$ source $S$*: KAN's per-edge learnable spline nonlinearity offers component-wise spectral adaptability, consistent with compact spectral structure reported for GS equilibria (e.g., Xie & Li, arXiv:2601.02942).
> > > - *Controlled evidence*: F-K and TKNO share the same KAN regressor; TKNO's gain isolates the trunk contribution on non-periodic GS boundaries, complementing the joint optimum in Table 4 (flux accuracy + PDE residual).
> > >
> > > **2. Empirical verification.**
> > > Using the same architecture under three training paradigms, we analyze latent geometry and KAN spectral diagnostics.
> > >
> > > *Latent space metrics.* Effective dimension $d_{\mathrm{eff}}=\exp(-\sum_i p_i\log p_i)$ (lower is more compact), and axis-LCFS Fisher ratio (higher is better separated):
> > >
> > > | Paradigm | $d_{\mathrm{eff}}$ (↓) | $S_{\mathrm{axis,LCFS}}$ (↑) |
> > > |:---|:---:|:---:|
> > > | physics-only | 2.09 | 47.1 |
> > > | **physics-anchored** | **1.57** | **160.0** |
> > > | data-only | 2.78 | 67.9 |
> > >
> > > Physics-anchored is the only one achieving both maximal compactness and strongest physically meaningful region separation. In parallel, KAN spectral analysis shows data-only concentrates excessive high-frequency energy near LCFS and is markedly rougher, while physics-anchored remains smooth and boundary-balanced.
> > >
> > > **3. OOD mechanism.**
> > > The two levels are consistent: physics-anchored yields compact structured latents + smooth spectrally efficient KAN functions; data-only yields boundary-concentrated sharp complexity. Since OOD shifts (shape/global/profile) perturb LCFS-related structure most, these artifacts fail first.
> > > Under local elliptic stability $\|\hat\psi-\psi^\ast\|\lesssim C\|\mathcal{R}\|$, lower PDE residual supports better OOD behavior on the correct branch. This matches observations: physics-anchored maintains much lower PDE residual and smaller OOD growth than data-only in the revised multi-shift evaluation.
> > >
> > > **4. Contribution beyond benchmarking.**
> > > (a) Identifying catastrophic OOD collapse in strongly nonlinear PDE surrogates and mitigating it via physics-anchored sparse supervision;
> > > (b) hypothesis-driven architecture design aligned with GS integral structure, validated by controlled comparison;
> > > (c) mechanism evidence (latent geometry + KAN spectral diagnostics) explaining *why* the paradigm works.
> > > Detailed tables (Chebyshev metrics, curvature/HF ratio, gradient-conflict analysis) are provided in our parallel response to Reviewer xPMp and the revised appendix.
> > >
> > > **Future directions.** Inspired in part by the reviewer's comments on evaluation scope and the fixed-boundary limitation, we plan to pursue:
> > >
> > > 1. **Free-boundary GSE**—coupling to vacuum fields and coil circuits, the natural next step from the fixed-boundary foundation.
> > > 2. **Richer distribution shifts**—wall conditions, impurity effects, and 3D perturbation proxies, extending the OOD stress-testing the reviewer encouraged.
> > > 3. **A foundation model for GS problems**—a unified pretrained model supporting forward solving, diagnostic-based reconstruction, profile inversion, and time-dependent equilibrium evolution, rather than training separate models per task.
> > > 4. **Physics-native architecture design**—moving beyond component selection toward structures with inductive biases tied to flux topology and LCFS geometry.
> > > 5. **Uncertainty quantification and OOD detection**—calibrating confidence on real device data so the surrogate can flag unreliable regimes.
> > >
> > > We are grateful that the reviewer's feedback helped shape both the current improvements and our future research direction.

---

### Decision · Program_Chairs · 2026-04-30

**Decision:**

Accept (regular)

**Comment:**

This paper considers the operator learning problem for nonlinear Grad-Shafranov Equation, and the main neural operator architecture is coupling Transformer and KAN (tested among other combinations using the common trunk-branch arch). After reading the paper myself, I would say this is a borderline paper. On one hand, it feels like your typical "applying neural operator with some incremental architectural mod to a certain PDE" paper (as mentioned by one reviewer). On the other hand, the problem itself is significant (real life data from EXL-50U tokamak) and worthy pursuing, which makes the OOD evaluation more relevant. Moreover, I found the negative results using unsupervised learning (neural operator trained merely on PDE residuals) are a worthy addition to the community.